

# Denitrification, dehydration and ozone loss during the Arctic winter 2015/2016

Farahnaz Khosrawi[1], Oliver Kirner[2], Björn-Martin Sinnhuber[1], Sören Johansson[1], Michael Höpfner[1], Michelle L. Santee[3], Lucien Froidevaux[3], Jörn Ungermann[4], Roland Ruhnke[1], Wolfgang Woiwode[1], Hermann Oelhaf[1], and Peter Braesicke[1]

[1]Institute of Meteorology and Climate Research, Karlsruhe Institute of Technology, Karlsruhe, Germany
[2]Steinbuch Centre for Computing, Karlsruhe Institute of Technology, Karlsruhe, Germany
[3]Jet Propulsion Laboratory, California Institute of Technology, California, USA
[4]Institute of Energy and Climate Research, Forschungszentrum Jülich, Jülich, Germany

*Correspondence to:* Farahnaz Khosrawi (farahnaz.khosrawi@kit.edu)

**Abstract.** The Arctic winter 2015/2016 was one of the coldest stratospheric winters in recent years. A stable vortex formed by early December and the early winter was exceptionally cold. Cold pool temperatures dropped below the Nitric Acid Tri-hydrate (NAT) existence temperature of about 195 K, thus allowing Polar Stratospheric Clouds (PSCs) to form. The low temperatures in the polar stratosphere persisted until early March allowing chlorine activation and catalytic ozone destruction.

Satellite observations indicate that sedimentation of PSC particles led to denitrification as well as dehydration of stratospheric layers. Model simulations of the Arctic winter 2015/2016 nudged toward European Center for Medium-Range Weather Forecasts (ECMWF) analyses data were performed with the atmospheric chemistry-climate model ECHAM5/MESSy Atmospheric Chemistry (EMAC) for the Polar Stratosphere in a Changing Climate (POLSTRACC) campaign. POLSTRACC is a High Altitude and LOng Range Research Aircraft (HALO) mission aimed at the investigation of the structure, composition and

evolution of the Arctic Upper Troposphere and Lower Stratosphere (UTLS). The chemical and physical processes involved in Arctic stratospheric ozone depletion, transport and mixing processes in the UTLS at high latitudes, polar stratospheric clouds as well as cirrus clouds are investigated. In this study an overview of the chemistry and dynamics of the Arctic winter 2015/2016 as simulated with EMAC is given. Further, chemical-dynamical processes such as denitrification, dehydration and ozone loss during the Arctic winter 2015/2016 are investigated. Comparisons to satellite observations by the Aura Microwave

Limb Sounder (Aura/MLS) as well as to airborne measurements with the Gimballed Limb Observer for Radiance Imaging of the Atmosphere (GLORIA) performed on board of HALO during the POLSTRACC campaign show that the EMAC simulations are in fairly good agreement with observations. We derive a maximum polar stratospheric $O_3$ loss of $\sim 2$ ppmv or 100 DU in terms of column in mid March. The stratosphere was denitrified by about 8 ppbv $HNO_3$ and dehydrated by about 1 ppmv $H_2O$ in mid to end of February. While ozone loss was quite strong, but not as strong as in 2010/2011, denitrification

and dehydration were so far the strongest observed in the Arctic stratosphere in the at least past 10 years.



# 1 Introduction

Since the early eighties, thus for more than 30 years, substantial ozone depletion has been observed each year during winter and spring in the Antarctic stratosphere (WMO, 2010). Polar ozone depletion is associated with enhanced chlorine from anthropogenic chlorofluorocarbons and heterogeneous chemistry under cold conditions. The deep Antarctic "hole" contrasts
with the generally weaker ozone depletions observed in the warmer Arctic (Solomon et al., 2014). Nevertheless, substantial ozone depletion has been observed for cold Arctic winters. Especially, in the past 15 years, ozone loss in the Arctic occasionally approached the degree of ozone loss in the Antarctic as e. g. in winter 2004/2005 (e.g. Manney et al., 2006; Tilmes et al., 2006; Livesey et al., 2015, and references therein) and 2010/2011 (e.g., Manney et al., 2011; Sinnhuber et al., 2011; Hommel et al., 2014).

Polar stratospheric clouds (PSC) play a key role in stratospheric ozone destruction in the polar regions (Solomon et al., 1986; Crutzen and Arnold, 1986). Heterogeneous reactions which take place on and within the PSC particles convert halogens from relatively inert reservoir species into forms which can destroy ozone in the polar spring (e.g., Peter, 1997; Solomon, 1999; Lowe and MacKenzie, 2008). PSCs form at altitudes between 15-30 km and consist of liquid and/or solid particles. According to their composition and physical state they have been classified into three different types: (1) supercooled ternary solutions
(STS), (2) Nitric Acid Trihydrate (NAT) and (3) ice. Liquid PSC cloud particles (STS) form by the condensation of water vapour ($H_2O$) and nitric acid ($HNO_3$) on the liquid stratospheric background sulfate aerosol particles at temperatures 2–3 K below the NAT existence temperature $T_{NAT}$ ($\sim 195$ K at 50 hPa) while for the formation of solid cloud particles (NAT and ice) lower temperatures are required (slightly above or below the ice frost point $T_{ice} \sim 188$ K at 50 hPa) (e.g. Carslaw et al., 1994; Koop et al., 1995).

Solid PSC particles can grow to larger sizes than liquid PSC particles and finally sediment out of the stratosphere (Fahey et al., 2001). The sedimentation of the solid particles can lead to dehydration and/or denitrification of the stratosphere. Solid $HNO_3$ containing PSC particles leading to denitrification can either consist of NAT or ice depending on the prevailing formation mechanism. It has been shown that the nucleation of NAT on ice is quite efficient (e.g., Fueglistaler et al., 2002; Hoyle et al., 2013). The sedimentation of large $HNO_3$ containing ice PSC particles can lead to greater denitrification than the sed-
imentation of (typically smaller) NAT or liquid PSC particles alone (Lowe and MacKenzie, 2008; Wohltmann et al., 2013; Manney and Lawrence, 2016).

Denitrification limits the deactivation process of the ozone destroying substances in springtime and thus leads to a prolongation of the ozone destroying cycles. Evidence of denitrification has been found in the Arctic and Antarctic from in situ and remote sensing observations (Fahey et al., 1990; Solomon, 1999; Waibel et al., 1999; Kondo et al., 2000; Santee et al., 2000;
Manney et al., 2011). Denitrification is most intense over the Antarctic region, where large fractions of available $NO_y$ are irreversibly removed from the stratosphere each winter. $NO_y$ is the sum of principal reactive nitrogen species, of which $HNO_3$, $NO$, $NO_2$, $N_2O_5$, and $ClONO_2$ are important in the lower stratosphere (Fahey et al., 1989). Dehydration in the stratosphere is generally observed over the Antarctic (e.g., Kelly et al., 1989; Vömel et al., 1995; Nedoluha et al., 2000) but only rarely in the Arctic (e.g., Fahey et al., 1990; Vömel et al., 1997; Pan et al., 2002; Schiller et al., 2002; Khaykin et al., 2013).





The Arctic winter 2015/2016 was one of the coldest stratospheric winters in recent years. A stable vortex formed already in early December and the early winter was exceptionally cold. The Arctic polar vortex in the early winter 2015/2016 was the strongest and coldest of the last 68 years (Matthias et al., 2016). Temperatures within the vortex dropped below the NAT existence temperature, thus allowing PSCs to form. Tropospheric and stratospheric cloud structures were observed simulta-
5 neously over Svalbard. Synoptic-scale polar stratospheric clouds extended over a nearly 8 km deep layer (Dörnbrack et al., 2016). The low temperatures in the polar stratosphere persisted until early March allowing PSC formation, chlorine activation and catalytic ozone destruction. Satellite observations indicate that sedimentation of PSC particles led to denitrification as well as dehydration of stratospheric layers (Manney and Lawrence, 2016). Widespread persistent ice PSC layers were observed by the Cloud-Aerosol Lidar and Infrared Pathfinder Satellite Observations (CALIPSO) (Voigt et al., 2016).

Model simulations of the Arctic winter 2015/2016 nudged toward European Center for Medium-Range Weather Forecasts (ECMWF) analyses were performed with the atmospheric chemistry-climate model ECHAM5/MESSy Atmospheric Chemistry (EMAC) for the POLSTRACC (Polar Stratosphere in a Changing Climate) campaign. POLSTRACC was a HALO mission (High Altitude and LOng Range Research Aircraft) aiming at the investigation of the structure, composition and evolution of the Arctic Upper Troposphere Lower Stratosphere (UTLS). The chemical and physical processes involved in Arctic strato-
spheric ozone depletion, transport and mixing processes in the UTLS at high latitudes, polar stratospheric clouds as well as cirrus clouds were investigated. In this study, an overview of the chemistry and dynamics of the Arctic winter 2015/2016 as simulated with EMAC is given. Chemical-dynamical processes such as denitrification, dehydration and ozone loss will be investigated and comparisons to satellite observations by the Aura Microwave Limb Sounder (Aura/MLS) as well as to airborne measurements with the Gimballed Limb Observer for Radiance Imaging of the Atmosphere (GLORIA) performed onboard of
HALO will be shown.

## 2 Model simulations and observations

### 2.1 EMAC

The ECHAM5/MESSy Atmospheric Chemistry (EMAC) model is a numerical chemistry and climate simulation system that includes sub-models describing tropospheric and middle atmosphere processes and their interaction with oceans, land and
25 human influences (Jöckel et al., 2010). It uses the second version of the Modular Earth Submodel System (MESSy2) to link multi-institutional computer codes. The core atmospheric model is the 5th generation European Centre Hamburg general circulation model (ECHAM5, Roeckner et al. (2006). For the present study we applied EMAC (ECHAM5 version 5.3.02, MESSy version 2.52) in T106L90MA and T42L90MA resolution, i.e., with a spherical truncation of T106 and T42 (corresponding to a quadratic Gaussian grid of approximately $1.125° \times 1.125°$ and $2.8° \times 2.8°$ degrees, respectively, in latitude and longitude)
with 90 vertical hybrid pressure levels from the surface up to $0.01\,\mathrm{hPa}$ (approx. $80\,\mathrm{km}$). A Newtonian relaxation technique of the prognostic variables temperature, vorticity, divergence and the (logarithm of the) surface pressure above the boundary layer and below $1\,\mathrm{hPa}$ towards ECMWF ERA-Interim reanalysis data (Dee et al., 2011) and ECMWF operational analysis was applied, respectively, in order to nudge the model dynamics towards the observed meteorology.



For the analyses of the Arctic winter 2015/2016 we use the EMAC data from a T106L90 simulation that was chemically initialised based on a former EMAC simulation. The T106L90 simulation was started on 1 July 2015 and continued until 30 April 2016, applying a nudging toward ECMWF operational analysis. For the comparisons to recent winters we performed an EMAC T42L90 simulation covering the time period 1 January 2008 to 30 April 2016. The T42L90 simulation was nudged

toward ECMWF ERA-interim analysis data until 30 June 2015 and toward ECMWF operational analysis data thereafter. In both simulations (T106L90 and T42L90) a comprehensive chemistry setup for the stratosphere and troposphere is included. Reaction rate coefficients for gas phase reactions and absorption cross sections for photolysis are taken from Atkinson et al. (2007) and Sander et al. (2011b). The applied model setup comprised among others the submodels: MECCA for the gas-phase chemistry (Sander et al., 2011a), JVAL for the calculation of photolysis rates (Sander et al., 2014), MSBM (Multi-phase

Stratospheric Box Model) for the processes related to polar stratospheric clouds (Kirner et al., 2011), TROPOP for diagnosing the tropopause and boundary layer height, SORBIT for sampling model data along sun-synchronous satellite orbits (Jöckel et al., 2010) as well as H2O for stratospheric water vapor.

## 2.2 Aura/MLS

The Microwave Limb Sounder (MLS) on the Earth Observing System Aura Satellite was launched in July 2004. The Aura/MLS

instrument is an advanced successor to the MLS instrument on the Upper Atmosphere Research satellite (UARS). MLS is a limb sounding instrument that measures the thermal emission at millimetre and submillimetre wavelengths using seven radiometers to cover five broad spectral regions (Waters et al., 2006). Measurements are performed from the surface to $90\,\mathrm{km}$ with a global latitude coverage from $82°\,\mathrm{S}$ to $82°\,\mathrm{N}$. Vertical profiles are measured every $165\,\mathrm{km}$ along the suborbital track with a horizontal resolution of $\sim$200-500 km along track and a footprint of $\sim$3-9 km across-track. Here, we use Aura/MLS version v4.2 $HNO_3$,

$O_3$ and ClO data. The data screening criteria given by Livesey et al. (2017) have been applied to the data.

A detailed assessment of the quality and reliability of the Aura/MLS v2.2 $HNO_3$ measurements can be found in Santee et al. (2007). The $HNO_3$ in v3.3 was significantly improved compared to v2.2. In particular, the low bias in the stratosphere was largely eliminated. Measurements of v4.2 $HNO_3$ are performed with a horizontal resolution of 400–500 km and a vertical resolution of 3–4 km over most of the vertical range. In the lower stratosphere, the precision has been estimated to be 0.6 ppbv

and the systematic uncertainty for $HNO_3$ is estimated to be $0.5-2$ ppbv (2-$\sigma$ estimates).

Detailed validation of the MLS $O_3$ v2.2 product and comparisons with other data sets can be found in Jiang et al. (2007), Froidevaux et al. (2008) and Livesey et al. (2008). In the stratosphere and above, v4.2 ozone profiles are very similar to the v2.2 and v3.3x/v3.4x profiles. Comparisons have indicated general agreement within 5–10 % with stratospheric profiles from satellite, balloon, aircraft, and ground-based data (Livesey et al., 2017).

The quality and reliability of the v2.2 MLS ClO measurements were assessed in detail by Santee et al. (2008). The ClO product was significantly improved in v3.3 and v3.4 (Livesey et al., 2013). In particular, the substantial ($\sim$ 0.1–0.4 ppbv) negative bias present in the v2.2 ClO values at pressures larger than 22 hPa was mitigated to a large extent, primarily through retrieval of $CH_3Cl$, which was a new MLS product in v3.3 and v3.4. The ClO retrieval is largely unchanged over much of the profile in v4.2. Measurements of ClO are performed with a horizontal resolution of 300-600 km and a vertical resolution





of 3-4.5 km. The precision lies generally within $\pm 0.1$ ppbv (Livesey et al., 2017). Although the negative bias at the lowest retrieval has not been entirely eliminated, we make no attempt to correct for it in this analyses.

## 2.3 GLORIA

The Gimballed Limb Observer for Radiance Imaging of the Atmosphere (GLORIA) combines a classical Fourier transform spectrometer with a 2-D detector array. The instrument takes limb images of the atmosphere from the flight altitude of HALO or M55-Geophysica down to 4 km. This results in vertical sampling steps of about 150 m at 8 km tangent height from a typical HALO flight level of 14 km. Individual images contain 128 pixels (spectra) in the vertical dimension and 48 pixels in the horizontal dimension. The spectra associated with the pixel rows are binned to reduce uncertainties. The spectral range of the observations currently extends from about 780 to 1400 cm$^{-1}$ (Riese et al., 2014). The list of species with signatures in this spectral range includes temperature, $H_2O$, HDO, $O_3$, $CH_4$, $N_2O$, CFC$-11$, CFC$-12$, HCFC$-12$, $SF_6$, $HNO_3$, $N_2O_5$, $ClONO_2$, $HO_2NO_2$, PAN, $C_2H_6$, $H_2CO$, $NH_3$. Details on the instrument design and calibration are given in Friedl-Vallon et al. (2014) and Kleinert et al. (2014). GLORIA is operated in a high-spectral, medium-spatial sampling ("chemistry") mode and a medium-spectral, high-spatial sampling ("dynamics") mode. The spectral samplings are 0.0625 cm$^{-1}$ for the chemistry mode and 0.625 cm$^{-1}$ for the dynamics mode (Riese et al., 2014). In this study, trace gas retrievals from measurements in the chemistry mode are used. A first validation of the retrieval results in the chemistry mode can be found in Woiwode et al. (2015).

## 3 Arctic winter 2015/2016

### 3.1 Overview

In the Arctic winter 2015/2016, temperatures were at record lows from December 2015 to early February 2016 with an unprecedented period of temperatures below the ice formation threshold (Manney and Lawrence, 2016). The extraordinarily strong and cold polar vortex in early winter (November-December 2015) was caused by very low planetary wave activity in the stratosphere (Matthias et al., 2016). The Arctic winter ended in early March by a major final sudden stratospheric warming. By mid-March, the vortex had been displaced far off the pole and split. The offspring vortices decayed rapidly, resulting in a full breakup of the vortex by early April (Manney and Lawrence, 2016).

In Figure 1 the temporal evolution of temperature and PSC surface area density at high latitudes (70-90°N) as function of pressure for the Arctic winter 2015/2016 (December 2015 to March 2016) as simulated with EMAC is shown. Temperatures below 195 K are found between 70 and 10 hPa from early December to end of January. Zonal mean temperatures remained cold afterwards, but not as cold as during December and January. Temperatures dropped during the first cold period (December to end of January) below the ice formation threshold temperatures (Manney and Lawrence, 2016) leading to unprecedented formation of ice PSCs. The simulated temperatures are in good agreement with observations from Aura/MLS (see Fig. 12 and Sect. 4.1).



The extensive formation of PSCs as simulated with EMAC can be seen in Figure 1 (bottom). Here, the total surface area density (liquid + solid) is shown. The first PSCs are found in the beginning of December and PSC formation maximises throughout January (between 80 and 20 hPa). During the second cold phase in February PSCs are still present but to a lesser extent. In Figure 2 the surface area densities of STS, NAT and ice as a function of pressure are shown (70-90°N). Since

the liquid particles have the largest surface area density $A_{STS}$ is almost identical to $A_{PSC}$. PSCs consisting of NAT are found between 150 and 20 hPa throughout December and January, and consisting of ice between 80 and 30 hPa in January. Compared to other extreme Arctic winters, e.g the 2010/2011 winter, much larger amounts of PSCs are simulated accordance with the preceding low temperatures for the Arctic winter 2015/2016. Furthermore, also the largest surface area density for ice is simulated for the Arctic winter 2015/2016 compared to previous Arctic winters e.g. the 2010/2011 Arctic winter, which has

been the most extreme in that respective so far (e.g., Manney et al., 2011; Sinnhuber et al., 2011; Hommel et al., 2014). Ice PSC persisted in 2015/2016 over a much longer time period than in e.g. the Arctic winter 2010/2011 as can be seen in the EMAC results for the Arctic winter 2010/2011 shown in Khosrawi et al. (2017).

## 3.2  Denitrification

Solid $HNO_3$ containing PSC particles can sediment out of the stratosphere and thus lead to an irreversible removal of $HNO_3$

(denitrification). Severe denitrification was observed by Aura/MLS in the Arctic winter 2015/2016. Figure 3 shows the $HNO_3$ gas phase distribution as simulated with EMAC for certain dates between 24 December 2015 and 12 February 2016 at 52 hPa. Strong gas phase removal of $HNO_3$ is evident throughout the entire period considered here. Gas phase $HNO_3$ is extremely low within the Arctic vortex in December and January, but mixing ratios increase somewhat (but still remain quite low) in February. That this gas phase removal of $HNO_3$ led to a permanent removal and thus to a denitrification of the stratosphere

can be seen from the redistribution of $NO_y$ in the model (Figure 4). In model simulations, denitrification can be quantified by applying a passive $NO_y^*$ tracer. Figure 4 shows the simulated $NO_y$ change averaged over 70-90°N as function of pressure and time. The unperturbed $NO_y^*$ was simulated by a passive tracer that was initialized according to the $NO_y$ distribution on 1 December 2015. The difference of $NO_y$ and $NO_y^*$ gives the amount of $NO_y$ that has been denitrified/re-nitrified ($\Delta NO_y$).

Figure 4 shows that strong denitrification is also simulated with EMAC for the Arctic winter 2015/2016. The maximum

denitrification is reached at the end of January (about 8 ppbv). Below the denitrified layers re-nitrification (about 4 ppb) due to the evaporation of the sedimenting PSC particles at lower pressure levels is clearly visible. Diabatic descent within the polar vortex causes the downward shift of the denitrified/re-nitrified areas. The mixing ratio increase at the re-nitrification altitudes is lower than the mixing ratio decrease at the denitrification altitudes, because the total mass of sedimented $HNO_3$ should be conserved and the pressure increases at decreasing altitude (Grooß et al., 2005).

## 3.3  Dehydration

The long period of temperatures below the ice formation threshold led to much greater dehydration than previously seen in the Arctic (Manney and Lawrence, 2016). Large areas of ice PSC throughout January were observed with CALIPSO that also were the greatest observed in the Arctic in the 8 years of the CALIPSO data record (Voigt et al., 2016). In the EMAC simulation large





areas of ice PSCs are simulated throughout January (Figure 2 bottom). Dehydration peaks in the EMAC simulation towards the end of January and is also the strongest simulated compared to other cold winters as e.g. the Arctic winter 2010/2011. The simulated dehydration in EMAC is also in agreement with observations. Trace gas measurements from Aura/MLS show that exceptional dehydration occurred during the Arctic winter 2015/2016 (Manney and Lawrence, 2016).

Figure 5 shows the EMAC $H_2O$ distribution at certain dates during the winter 2015/2016 at 52 hPa. On 24 December 2015 the $H_2O$ distribution shows the usual background $H_2O$ mixing ratios in the Arctic region. From January onwards, mixing ratios drop and an area with mixing ratios below 5 ppmv is found north of Scandinavia. Mixing ratios decrease further throughout January and the area of dehydration increases. From February onwards $H_2O$ mixing ratios start to increase again, but still remain lower than the pre-winter values.

Dehydration from the EMAC simulation is derived by using total stratospheric hydrogen ($2CH_4+H_2O$) as substitute for a passive $H_2O$ tracer (e.g., Rinsland et al., 1996; Schiller et al., 1996). The exceptional dehydration during the Arctic winter 2015/2016 can be seen in the temporal evolution of $\Delta H_2O$ as function of pressure averaged over 70-90°N (Figure 6). The decrease of $\Delta H_2O$ throughout January and February shows dehydration of the lower stratosphere of around 1 ppmv extending between 60 and 30 hPa. Dehydration reaches its maximum in mid January ($\Delta H_2O$ of up to 2 ppmv). Below the dehydrated

areas re-hydration (up to 0.6 ppmv) due to the evaporation of the sedimenting PSC particles at larger pressure levels (lower altitudes) is clearly visible.

### 3.4 Ozone loss

The Arctic winter 2015/2016 had the greatest potential yet seen for record Arctic ozone loss since temperatures in the Arctic lower stratosphere were at record lows from December 2015 to early February 2016 (Manney and Lawrence, 2016; Matthias

et al., 2016). Ozone destruction began earlier and proceeded more rapidly than in 2010/2011, the winter that so far has been the one with the strongest observed ozone loss in the Arctic (Manney et al., 2011). However, a major final stratospheric warming in early March 2016 lead to a vortex split and a full breakdown of the vortex by early April (Manney and Lawrence, 2016).

In the following the EMAC simulation is used to investigate ozone depletion during the Arctic winter 2015/2016 and compare the results with previous Arctic winters. Ozone depletion ($\Delta O_3$) from the model simulation is determined by the difference

between the modelled ozone ($O_3$) and an artificial passive ozone tracer ($O_3^*$). The passive ozone tracer was initialised on 1 December 2015 according to the ozone distribution on that day and was then advected and mixed as all other chemical species but did not undergo any chemical changes. The simulated ozone depletion (averaged over 70-90°N) is shown in Figure 7. From mid January onwards ozone depletion is visible in the EMAC simulation and a maximum depletion of about 2.1 ppmv is reached at about 30 hPa in mid March.

The simulated total column ozone loss time series from 1 December to 31 March averaged over 70-90°N is shown in Figure 8. Changes in the total column become visible from the end of January onwards. The absolute maximum in total column ozone loss of about 120 DU is reached on 7 March.





### 3.5 Comparison to recent Arctic winters

For the comparison of the EMAC simulation of the Arctic winter 2015/2016 to previous Arctic winters the EMAC T42L90 simulation is used. The results from both simulations, T42L90 and T106L90, are quite similar as can be seen from the time series comparison shown in Section 4 where the EMAC simulations are compared to Aura/MLS observations. The agreement with the Aura/MLS measurements is slightly better for the T106L90 simulation.

Although considerable ozone loss occurred during the Arctic winter 2015/2016, ozone loss was not as strong as in 2010/2011 as can be seen from Figure 9 and Figure 10. In Figure 9 the March mean $O_3$ column is shown for the years 2010 to 2016. Very low $O_3$ column values are found in March 2011. Column values reach 250 DU. In March 2016, however, the $O_3$ column remains quite high.

Figure 10 shows the Arctic mean column $O_3$ time series (averaged over 60° to 90° N) from 1 December to 30 April for the four Arctic winters 2009/2010, 2010/2011, 2013/2014 and 2015/2016. The EMAC Arctic mean column shows considerable interannual variability. In contrast to the other Arctic winters very low $O_3$ is found in 2010/2011. The extreme low $O_3$ column that we find in the EMAC simulation for the winter 2010/2011 is in agreement with the results from Strahan et al. (2013) and Manney et al. (2011) using observations and model simulations. In 2015/2016 the $O_3$ column was comparably low in early winter, but from February onwards the $O_3$ column started to increase significantly due to the disturbances of the Arctic stratosphere by sudden stratospheric warmings. In fact, winters with above average stratospheric wave activity have a warm, disturbed vortex, while winters with weak wave driving have a cold, long lasting vortex, with well-known impacts on Arctic March temperatures and $O_3$ column (Strahan et al. (2013) and references therein). Manney and Lawrence (2016) found from MLS observations that ozone continued to decrease in the vortex at a rate slightly faster than that in 2011 until the beginning of March 2016. However, around mid-March ozone increased for the rest of the winter so that the ozone values always remained higher than in 2011. This is also seen in the EMAC simulation. Therefore, our model simulations are in agreement with the results by Manney and Lawrence (2016) that in the Arctic winter 2015/2016 the stratosphere had the greatest potential yet seen for a massive Arctic ozone loss due to record low temperatures, but was disrupted by the final sudden warming in early March.

On the other hand, denitrification and dehydration were the strongest observed so far (Manney and Lawrence, 2016). From the EMAC simulation the same result as from the observations is derived. Figure 11 shows the time series of $HNO_3$ and $H_2O$ for the same four Arctic winters as shown in Figure 10. At 48 hPa several ppbv lower $HNO_3$ mixing ratios than in previous cold Arctic winters is found from December to February. Pre-winter $HNO_3$ mixing ratios were around 11 ppbv and drop to 4 ppbv in mid January. How much lower the $H_2O$ mixing ratios drop due to the dehydration during the Arctic winter 2015/2016 compared to other Arctic winters can be seen in the $H_2O$ time series at 48 hPa (Figure 11 bottom). In early December, $H_2O$ mixing ratios are as high as 5.8 ppmv and decrease to 5.2 ppmv, but decrease for a short period towards the end of January to even lower values (4.7 ppmv). From the end of January the $H_2O$ mixing ratios increase slowly, but still remain lower than the pre-winter values. The $H_2O$ mixing ratios are in addition ∼1-1.5 ppmv lower in January and February than in previous cold Arctic winters.





## 4   Comparison to observations

In this study, we compare the EMAC simulations for the Arctic winter 2015/2016 to Aura/MLS observations. In another study Khosrawi et al. (2017) the EMAC simulations of $HNO_3$, temperature and PSC volume density were compared for the Arctic winters 2009/2010 and 2010/2011 with satellite observations (Envisat/MIPAS and Aura/MLS). Here, we consider in addition

to temperature and $HNO_3$ other trace gases such as $O_3$, ClO (section 4.1) as well as $H_2O$ and compare the simulations to Aura/MLS observations. Additionally, the EMAC simulations are compared to remote sensing observations from GLORIA performed during the POLSTRACC measurement campaign (section 4.2). For the comparisons to Aura/MLS the EMAC SORBIT ouput is used (Jöckel et al., 2010) while for the comparison to GLORIA the EMAC global field output is interpolated to the GLORIA measurement geolocations.

### 4.1   Comparison to Aura/MLS

Figure 12 shows a comparison of the temperature, $HNO_3$, and $O_3$ distribution measured by Aura/MLS with the ones simulated with EMAC at about 50 hPa on 15 January 2016. For temperature as well as for $HNO_3$ and $O_3$ the simulations are in general agreement with the Aura/MLS observations. Nevertheless, some differences are found between model simulations and observations. Temperatures as simulated with EMAC tend to be slightly warmer than measured outside the polar vortex. The trace

gas distributions of $HNO_3$ and $O_3$ simulated with EMAC show more fine-scale structures which may be related to the higher horizontal resolution ($1.125° \times 1.125° \sim 125\,km \times 125\,km$ or less dependent on latitude) of the EMAC simulation compared to Aura/MLS (measurements every $1.5° \sim 165\,km$ and resolution of 200-500 km along track). Generally, the simulated $HNO_3$ mixing ratios are slightly lower than the ones measured with Aura/MLS while the simulated $O_3$ mixing ratios are quite similar to the observed $O_3$.

The temporal development of ClO, $HNO_3$ and $O_3$ averaged over 70-90°N during the Arctic winter 2015/2016 as function of pressure as simulated with EMAC and observed by Aura/MLS is shown in Figure 13. Here, the EMAC SORBIT output is used (Jöckel et al., 2010) where EMAC is sampled along the sun-synchronous orbit of Aura/MLS. The use of the SORBIT output improves the agreement between observations and simulations of trace gases with a diurnal cycle as e. g. ClO significantly, but has a rather minor impact on the comparison between observations and simulations for other trace gases as e. g. $O_3$. Generally,

the temporal evolution of the trace gas distributions is realistically reproduced in the EMAC simulation. Nevertheless, there are some differences found between measurement and model simulations. In the observations, increased ClO mixing ratios are already found in December whereas in the model simulation the increase of ClO occurs somewhat later. However, the enhancement of $ClO_x$ in the EMAC simulation is found at the same time as in the Aura/MLS ClO observation, thus indicating that the later increase in ClO is probably not caused by the activation of chlorine being too late in the model simulation but

rather of the partitioning between the active chlorine species. The ClO mixing ratios are maximum in February in both the observations and model simulations, but at the maximum higher mixing ratios are found and these extend over a larger vertical range in the EMAC simulation than in Aura/MLS observations.





The temporal evolution of the $HNO_3$ distribution as a function of pressure shows that the model simulation captures the general features well. In early December $HNO_3$ mixing ratios are slightly underestimated by EMAC ($\sim$1 ppbv). Gas phase removal of $HNO_3$ due to uptake in PSCs is more strongly simulated at higher pressure levels (Dec to Jan at around 100 hPa) while underestimated at lower pressure levels (January to February at around 50 hPa). PSCs composed of NAT form in EMAC as soon as temperatures drop below $T_{NAT}$-3 K which results often in a too early formation of NAT particles. Among other things, this has also an impact on the denitrification as was found in another study comparing EMAC simulations for the Arctic winter 2009/2010 and 2010/2011 with Envisat/MIPAS and Aura/MLS observations (Khosrawi et al., 2017). Because NAT is calculated before STS in the model, the NAT formation occurs at the expense of STS since the available $HNO_3$ is first consumed by the NAT clouds (e.g., Wohltmann et al., 2013). Since in reality STS and NAT clouds are often observed at the same time (e.g., Pitts et al., 2011; Peter and Grooß, 2012), this could be one explanation for the deviations in the $HNO_3$ distribution.

The temporal evolution of EMAC $O_3$ (Figure 13 bottom panel) is quite similar to that observed by Aura/MLS, especially in the lower stratosphere. In the upper stratosphere more $O_3$ is brought down leading to higher $O_3$ in the EMAC simulation above 20 hPa in March compared to Aura/MLS.

Figure 14 shows the time series of $HNO_3$ and $O_3$ at 50 hPa for Aura/MLS and the EMAC T42 and T106 simulation. In the EMAC simulation the $HNO_3$ is slightly underestimated in the beginning of December (by about 1 ppbv). Larger differences at 50 hPa are found in the time period of denitrification (end of December to end of January). At this time, the EMAC simulations underestimate denitrification at 50 hPa by about 2-3 ppbv. The simulated $O_3$ is in good agreement with Aura/MLS measurements during December and January. From February onwards the simulated $O_3$ is up to 0.25 ppmv higher than the observed $O_3$. For both species the T106L90 simulation agrees slightly better with the Aura/MLS observations. How important the resolution of the model simulation is and that especially along the HALO flight tracks a better agreement with measurements is derived with the EMAC T106L90 resolution is shown in Sinnhuber et al. (2017).

## 4.2 Comparison to GLORIA

The EMAC simulations were performed in support of the POLSTRACC campaign. This allows as to evaluate the model performance in the lower stratosphere by comparison to high resolved measurements performed onboard HALO. Here, we show a comparison of EMAC $HNO_3$ and $O_3$ to the remote sensing instrument GLORIA. Further comparisons of EMAC to GLORIA for several trace gases will be shown in Johansson et al. (2017) and Braun et al. (2017) and comparison to in-situ instruments onboard HALO are shown in Sinnhuber et al. (2017).

The comparison shown here is for the POLSTRACC flight 21 on 18 March 2016. EMAC output has been taken along the times and location of GLORIA measurements (Figure 15). The GLORIA measurements in chemistry mode of flight 21 used in this comparison were performed over Scandinavia. By mid-March the polar vortex had been displaced off the pole and split. The colder offspring vortex was centered over Northern Russia and during flight 21 air masses at the border of this offspring vortex have been probed.





EMAC $HNO_3$ and $O_3$ compares generally well to GLORIA in terms of the distribution and mixing ratios. However, at 12-14 km, the area where the polar vortex has been probed, $O_3$ mixing ratios from EMAC are slightly lower than the ones observed by GLORIA. The same holds for $HNO_3$, but differences between EMAC and GLORIA are larger. The underestimation of polar vortex $O_3$ in the EMAC simulation could be either caused by a too weak downward transport or a too strong ozone destruction

in the model. The former reason, however, is more likely, since a well known feature in EMAC is that the downward transport is underestimated in the lower parts of the polar vortices (Brühl et al., 2007). Further, ozone loss in EMAC is rather underestimated than overestimated as was found in the evaluation study by Khosrawi et al. (2009).

Another difference between EMAC and GLORIA is that less fine-scale structure is simulated with EMAC than observed by GLORIA, which is probably due to the rather coarse horizontal resolution of EMAC (T106 corresponding to $1.125° \times 1.125°$)

compared to GLORIA. Nevertheless, this comparison and the ones presented in (Johansson et al., 2017) show that EMAC simulations can be used for comparisons to aircraft measurements. In the future simulations with EMAC with an even higher horizontal resolution (T255) are anticipated which are expected to result in even better agreement with observations derived onboard aircraft.

## 5   Conclusions

In this study, an overview of the chemistry and dynamics of the Arctic winter 2015/2016 as simulated with EMAC was given. Chemical-dynamical processes such as denitrification, dehydration and ozone loss were investigated and comparisons to satellite observations by the Aura/MLS as well as to airborne measurements with GLORIA performed onboard of HALO were shown.

From the EMAC simulation we derive a maximum polar stratospheric $O_3$ loss of $\sim 2$ ppmv or 100 DU in terms of column

in mid March. The stratosphere was denitrified by about 8 ppbv $HNO_3$ and dehydrated by about 1 ppmv $H_2O$ in mid to end of February. In agreement with the analyses of Aura/MLS observations by Manney and Lawrence (2016) we find that ozone loss was quite strong in 2015/2016, but not as strong as in 2010/2011. Denitrification and dehydration on the other hand were so far the strongest observed in the Arctic stratosphere.

Comparison of trace gas distributions of $HNO_3$, ClO and $O_3$ shows that the EMAC simulations generally reproduce well the

Aura/MLS observations during the Arctic winter 2015/2016. However, there are some differences between the EMAC simulations and observations which need sensitivity studies in the future to improve the agreement between the model simulations and observations. In the EMAC simulation the $HNO_3$ is slightly underestimated (by about 1 ppbv). Larger differences are found in the area of denitrification which could be related to the partitioning between STS and NAT in the model. The observed increase in ClO at the beginning of the winter is simulated later with EMAC. Considering ClOx we found that activation of chlorine

occurs in EMAC at the same time as in the observations and that the difference in ClO is therefore probably rather caused by the partitioning of the chlorine species than by a too late activation of chlorine.




The comparison to GLORIA measurements shows that EMAC can reproduce the observations. Further, this comparison shows that, though EMAC is a climate model, EMAC simulations can be applied in support of aircraft campaigns and that these simulations provide a valuable data set not only for flight analyses but also for measurement - model intercomparisons.

*Acknowledgements.* We would like to thank the European Centre for Medium-Range Weather Forecasts (ECMWF) for providing their
5    meteorological analyses. We also would like to thank the MLS team for providing their data. MLS data were obtained from the NASA Goddard Earth Sciences and Information Center. Work at the Jet Propulsion Laboratory, California, Institute of Technology, was done under contract with the National Aeronautics and Space Administration. S. Johansson has received funding from the European Community's Seventh Framework Programme (FP7/2007-2013) under grant agreement 603557. We would like to thank the GLORIA team for performing the measurements onboard HALO during the POLSTRACC campaign. Atmospheric research with HALO is supported by the Priority
10    Programme 1294 of the Deutsche Forschungsgemeinschaft. EMAC simulations were performed on the Institute Cluster II at the Steinbuch Center for Computing at Karlsruhe Instititute of Technology. We acknowledge support by Deutsche Forschungsgemeinschaft and Open Access Publishing Fund of Karlsruhe Institute of Technology.




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



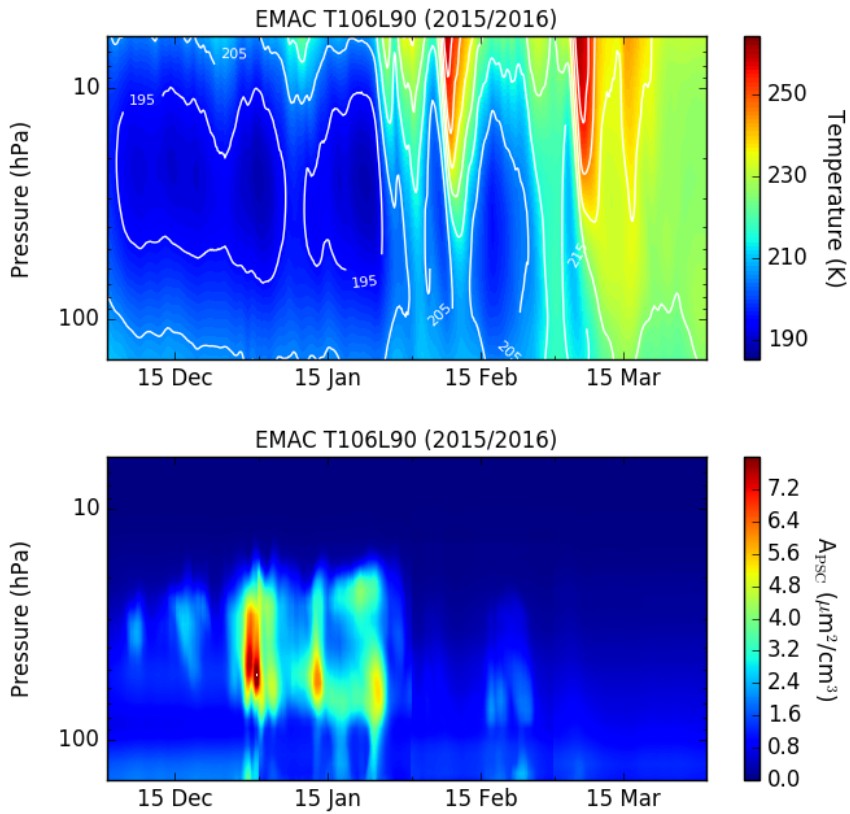

**Figure 1.** Temporal evolution of temperature and surface area density of PSC particles (liquid + solid) at northern high latitudes (70-90°N) as function of pressure during the Arctic winter 2015/2016 as simulated with EMAC T106L90.



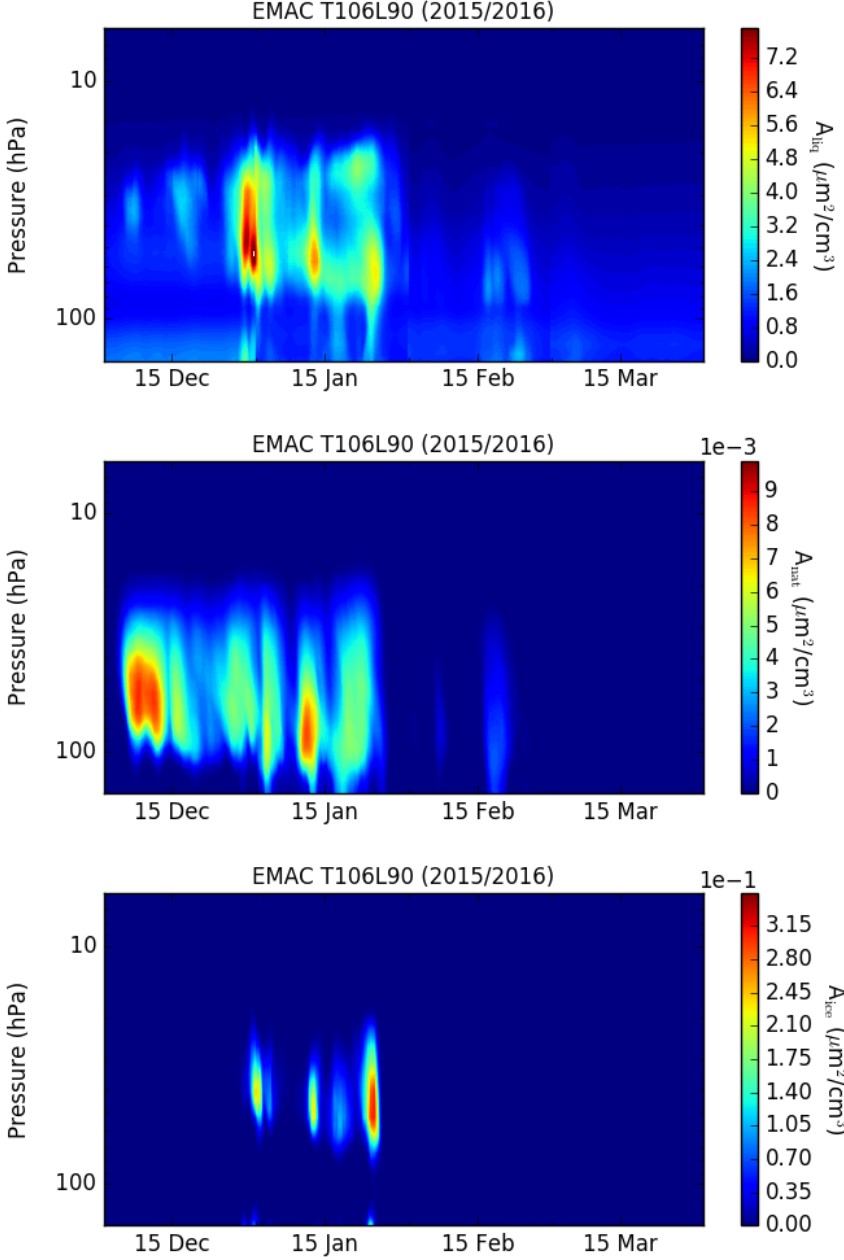

**Figure 2.** Temporal evolution of surface area density of STS (top), NAT (middle) and ice (bottom) particles at northern high latitudes (70-90°N) as function of pressure during the Arctic winter 2015/2016 as simulated with EMAC T106L90. Note the differences in the color bar for $A_{\mathrm{liq}}$ ($\mu$m$^2$/cm$^3$), $A_{\mathrm{NAT}}$ ($10^{-3}$ $\mu$m$^2$/cm$^3$) and $A_{\mathrm{ice}}$ ($10^{-1}$ $\mu$m$^2$/cm$^3$).




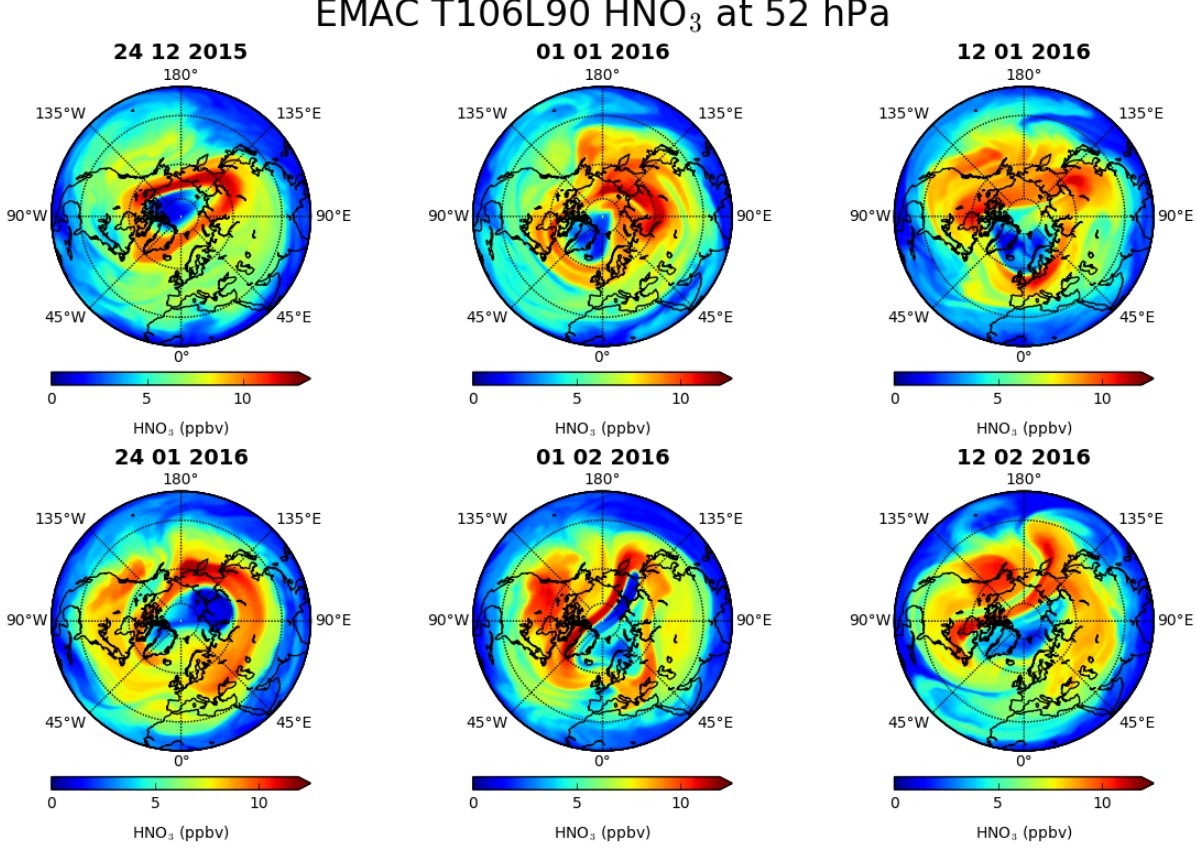

**Figure 3.** Distribution of $HNO_3$ as simulated with EMAC T106L90 at 52 hPa on certain dates between 24 December 2015 and 12 February 2016.





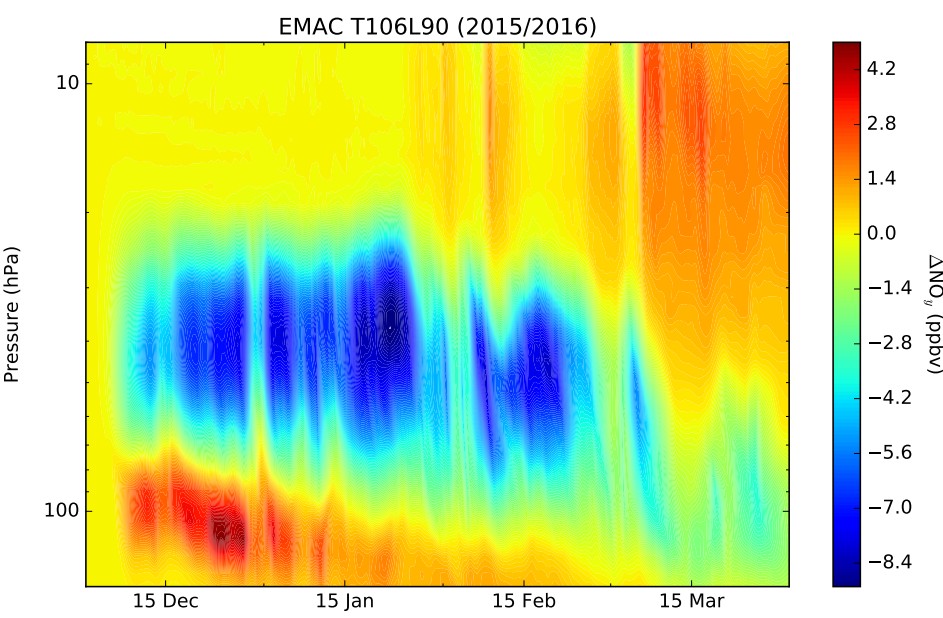

**Figure 4.** Redistribution of NOy simulated with EMAC T106L90 (difference of NOy and the passive tracer $NO_y^*$, averaged over 70-90°N).



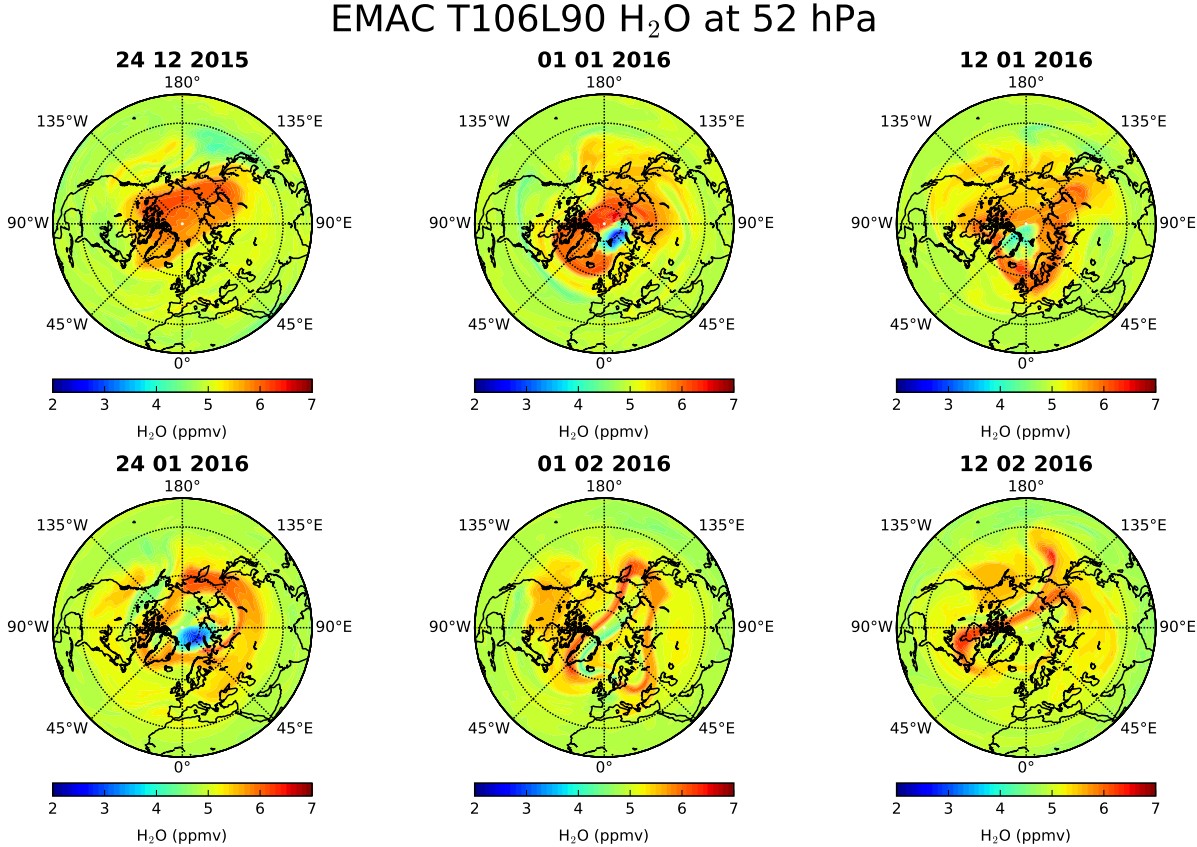

**Figure 5.** Distribution of $H_2O$ as simulated with EMAC T106L90 at 52hPa on certain dates between 24 December 2015 and 12 February 2016.





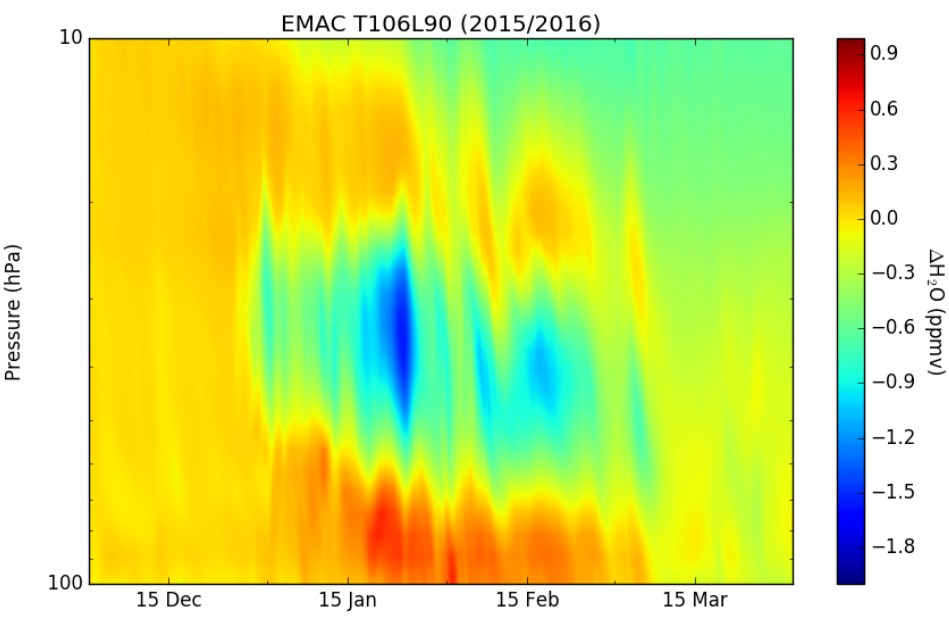

**Figure 6.** Redistribution of $H_2O$ as simulated with EMAC T106L90 at northern high latitudes (70-90°N) as function of pressure during the Arctic winter 2015/2016 (difference of total hydrogen ($2CH_4+H_2O$) at time t and total hydrogen at time $t_0$ (1 December)).





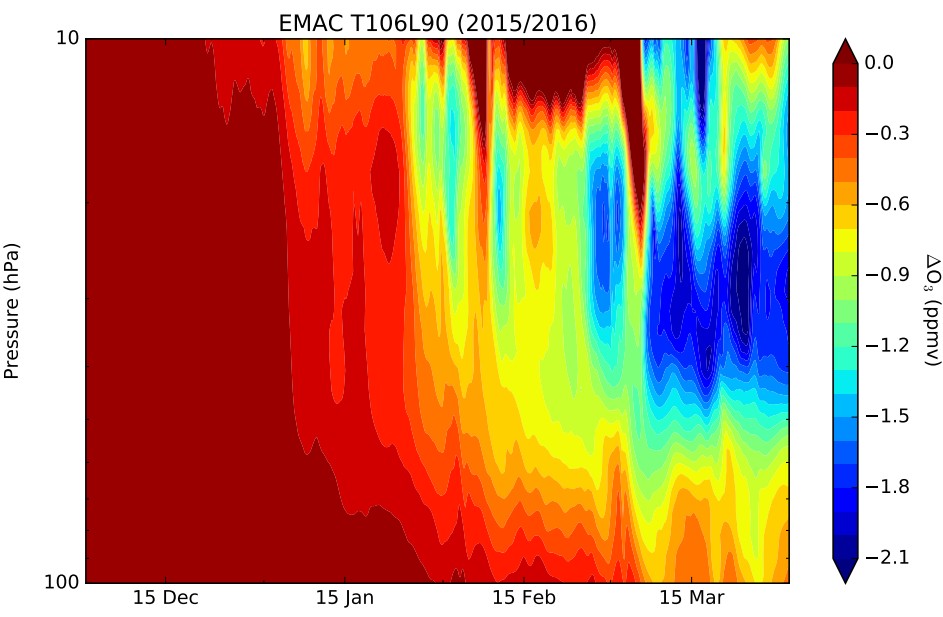

**Figure 7.** Ozone loss as simulated with EMAC T106L90 (difference of $O_3$ and the passive tracer $O_3^*$, averaged over 70-90°N) as function of time and pressure for the Arctic winter 2015/2016.





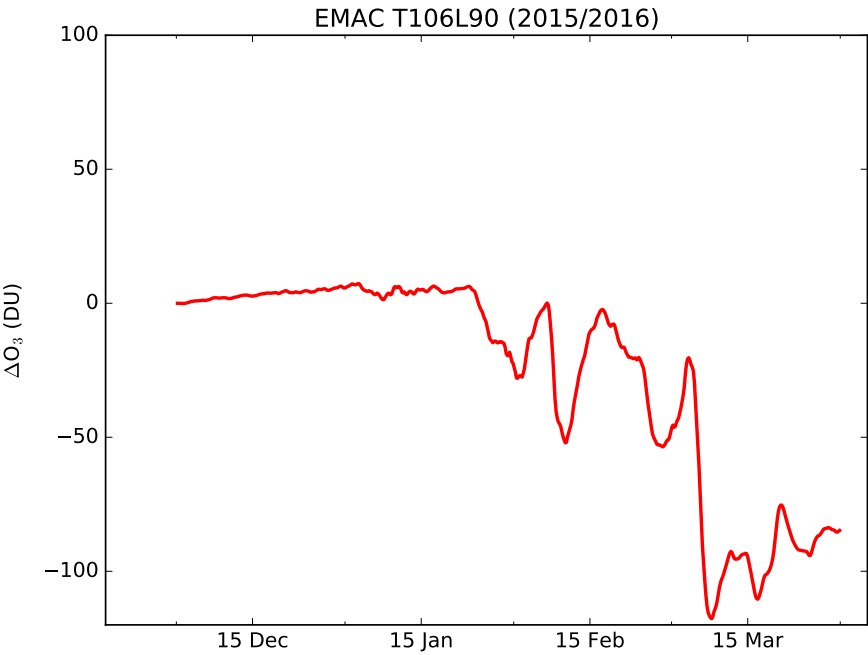

**Figure 8.** Total column ozone loss ($\Delta O_3$) from EMAC T106L90 (70-90°N) for the Arctic winter 2015/2016. Total column loss has been derived from the difference between the active tracer $O_3$ and the passive tracer $O_3^*$.





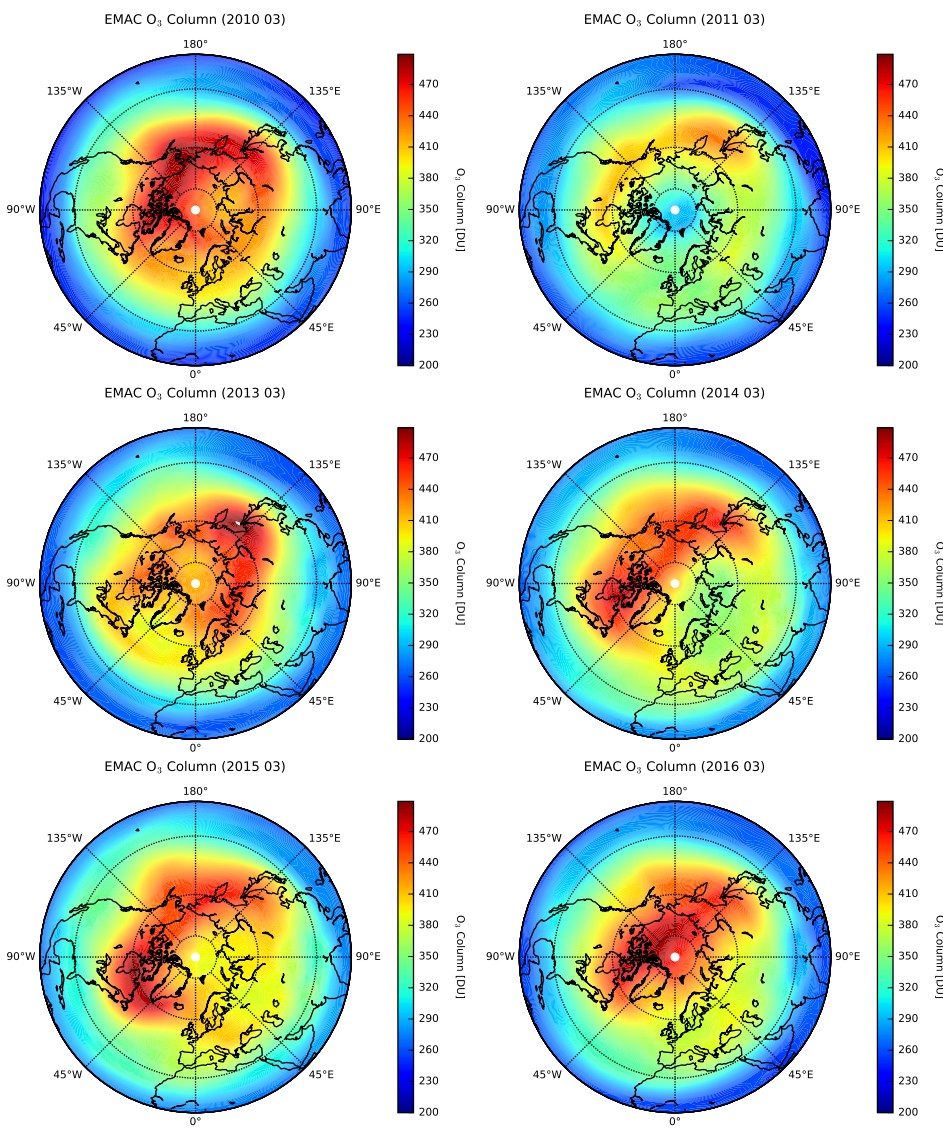

**Figure 9.** Total ozone column (March monthly mean) from EMAC for the years 2010-2016 (Results from the EMAC T42L90 Simulation are shown here).



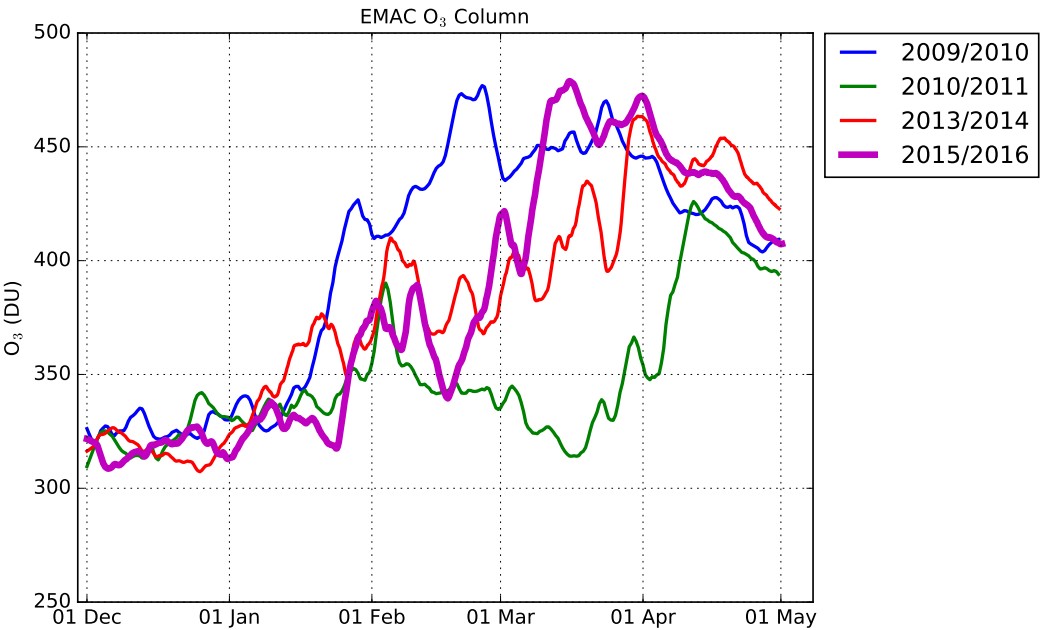

**Figure 10.** Ozone ($O_3$) column time series for the Arctic winters 2009/2010 (blue), 2010/2011 (green), 2013/2014 (red) and 2015/2016 (magenta) averaged over 60-90°N (Results from the EMAC T42L90 Simulation are shown here).





**Figure 11.** Tracer time series of $HNO_3$ and $H_2O$ for the Arctic winters 2009/2010 (blue), 2010/2011 (green), 2013/2014 (red) and 2015/2016 (magenta) at 48 hPa averaged over 70-90°N (Result from the T42L90 Simulation are shown here).





**Figure 12.** Temperature, HNO$_3$, O$_3$ distribution measured by Aura/MLS (left) and simulated by EMAC T106L90 (right) at ∼50 hPa on 15 January 2016.





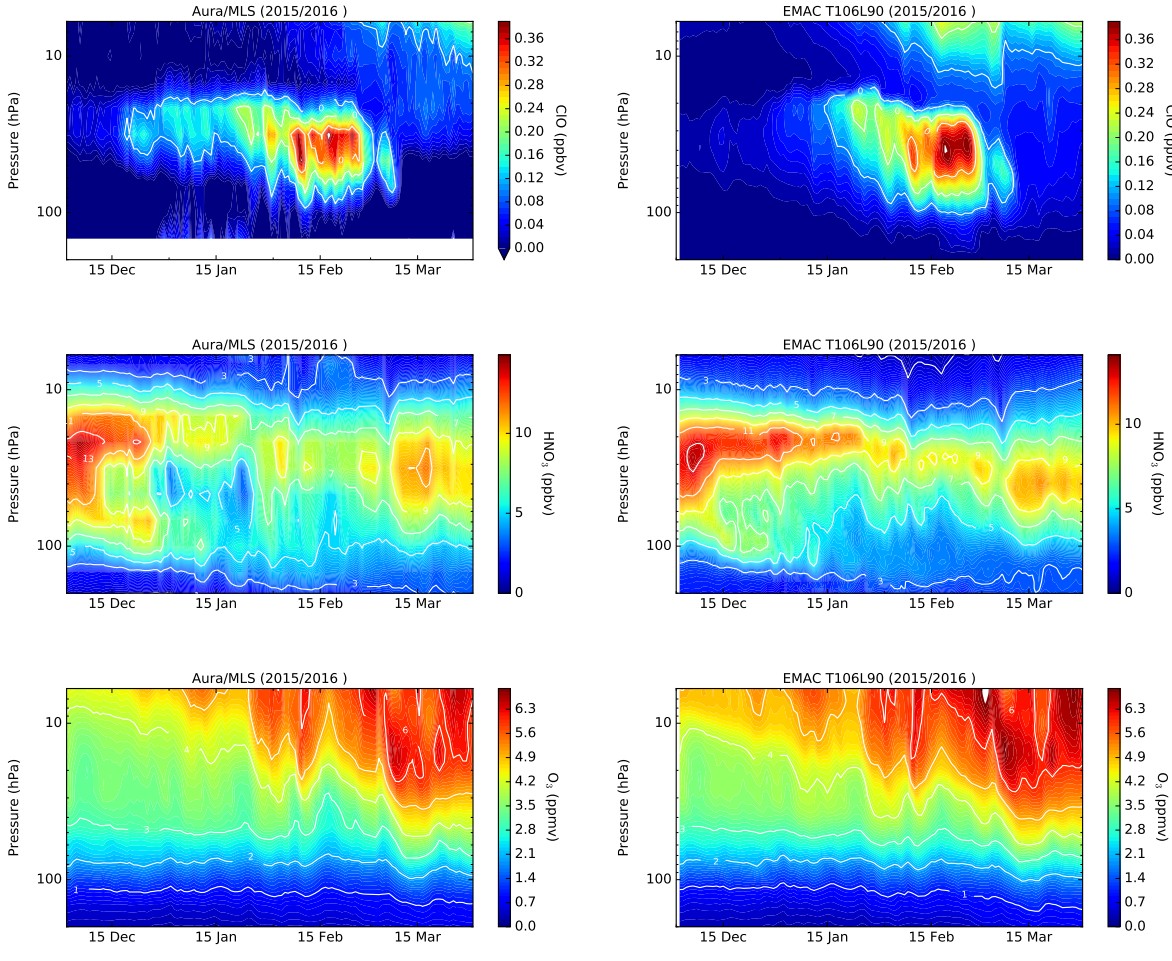

**Figure 13.** Temporal evolution of daily mean ClO, HNO₃ and O₃ at northern high latitudes (averaged over 70-90°N) as function of pressure as observed by Aura/MLS (left) and simulated by EMAC T106L90 (right) for the Arctic winter 2015/2016 (EMAC SORBIT output used).





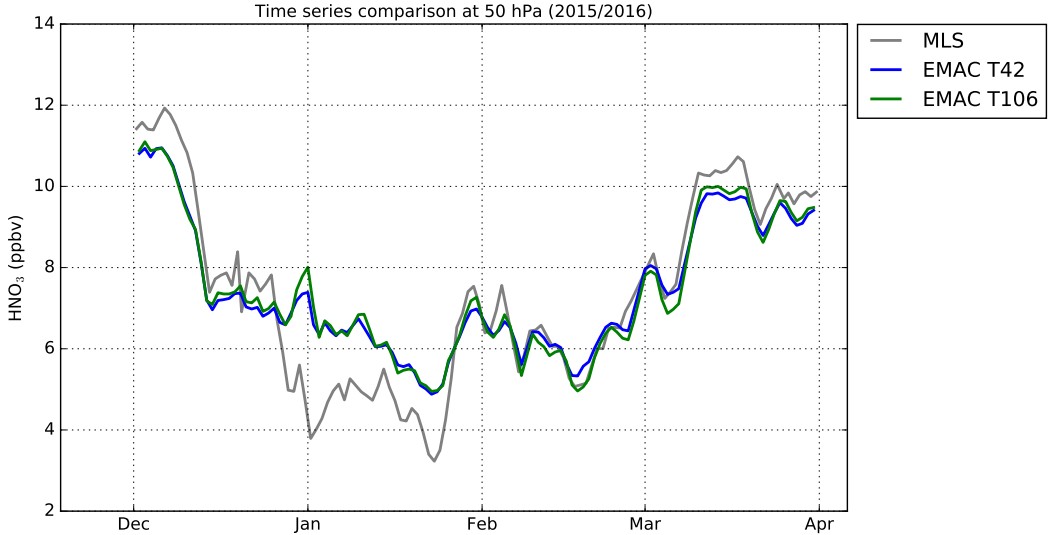

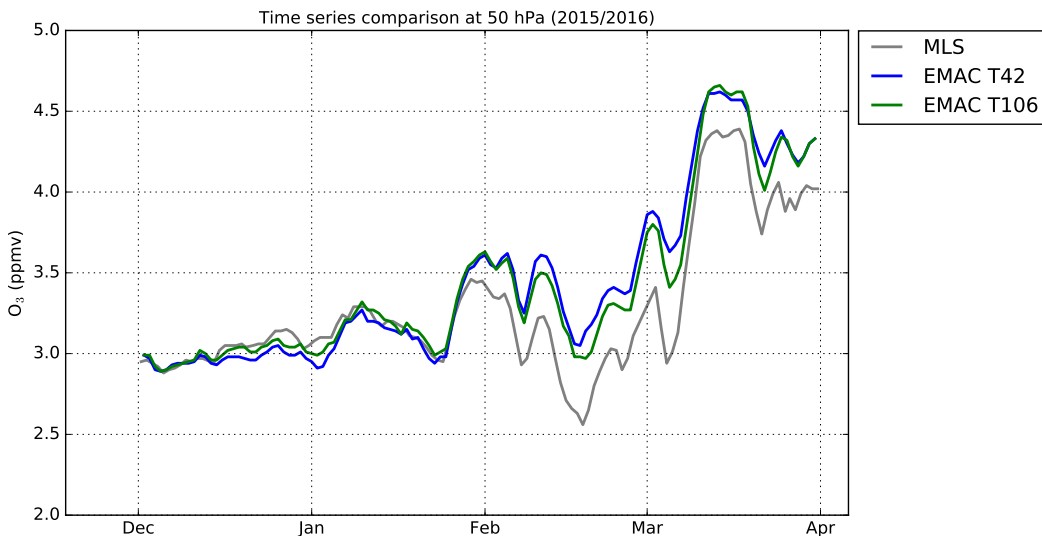

**Figure 14.** Time series of $HNO_3$ and $O_3$ from Aura/MLS measurements (grey) and from the EMAC T42L90 (blue), EMAC T106L90 (green) at $\sim$50 hPa averaged over 70-90°N (EMAC SORBIT output used).



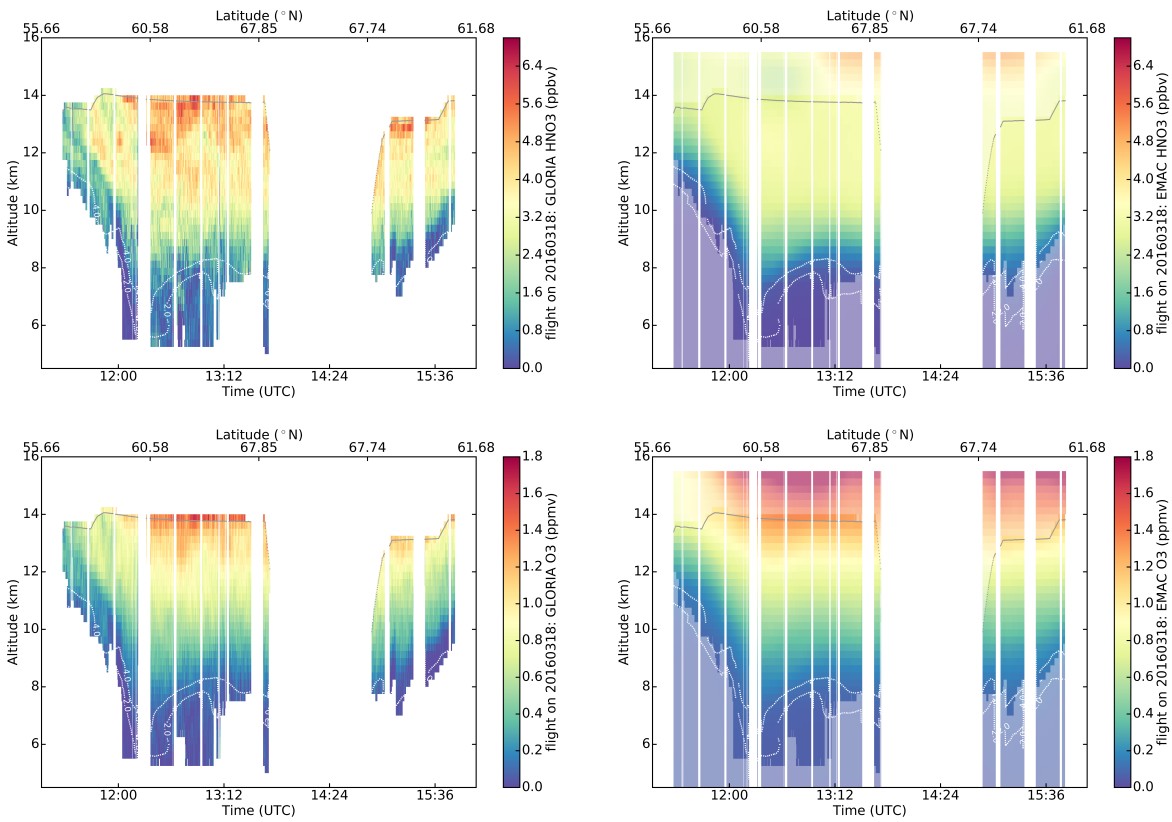

**Figure 15.** GLORIA $HNO_3$ and $O_3$ observations during flight 21 on 18 March 2016 (left) and EMAC T106L90 output along the flight track (right).