# Peer review of "Denitrification, dehydration and ozone loss during the Arctic winter 2015/2016"

_Atmospheric Chemistry and Physics, 2017_

## Referee Comment (RC1) · Anonymous Referee #1 · 30 Jul 2017

General statement: This is a fine paper on an important topic that merits publication after revision. I do have several comments for the authors, delineated below. Important ones are marked with *.

*1) The question of how denitrification and dehydration as such, versus a longer duration of cold temperatures into later parts of the spring season, have not been examined quantitatively here. The authors should therefore avoid trying to make statements about how important denitrification and dehydration were (or would be) for the ozone loss. I suggest that the authors consider this point carefully in revision. I point out one place to make a change in text but I think there could well be others.

[Figure]

2) page 2, line 15. Please change 'ice' to 'water ice' here since some literature speaks of nitric acid ices. With this change, I don't think you need to say 'water ice' later in the text; doing it once is sufficient.

*3) page 2, line 27. This statement makes a lot of assumptions that I don't think are merited. First, it ignores the literature on 'denoxification', much of which suggests that denoxification later in the spring, when there is more sunlight, can be as important or more so in prolonging ozone loss provided temperatures are cold enough. Second (and related), I would argue that prolonging the ozone loss depends more on vortex stability and dynamics than it does on the degree of denitrification. Please add a discussion of these issues here, with appropriate references.

4) page 4, line 28-29. I don't think these accuracy claims are true for MLS below 100 mb. Please check.

5) page 5, line 2. Missing a word. Lowest retrieval level?

6) page 5, line 30. Reader needs a pointer ahead to indicate that you will define what you mean by 'unprecedented'. Add 'leading to unprecedented formation of ice PSCs (defined quantitatively below)'. . ..

7) page 7, line 10. 2CH4+H2O isn't quite total hydrogen. I don't think it matters much for your purposes, but please have a look at LeTexier et al. (QJRMS, 1988) on this.

*8) page 7, line 18, 19. Need to be more careful here. You could say something like 'The Arctic winter 2015/2016 had the greatest potential yet seen for record Arctic ozone loss if the vortex had remained stable (and temperatures had therefore remained cold) through late March'.

*9) page 9, line 30. Interesting – can you say something more about which other ClOx species are likely to be holding too much active chlorine? Cl2O2? ClONO2? Also, I don't think you can rule out that the activation is at the right time but just too weak? What is your justification for ruling that out? Please clarify this here, as well as in other

places where it is mentioned.

---

## Referee Comment (RC2) · Anonymous Referee #2 · 31 Jul 2017

Khosrawi et al. present a detailed analysis of polar processes occurring at high northern latitudes during the Arctic winter 2015/16. In particular, they compare simulations carried out with a nudged version of the EMAC CCM with a range of satellite and aircraft observations. The analysis presented in the paper is of high standard and explores an important and relevant topic within the scope of ACP and as such merits publication following revision. I have several comments the authors should address before publication:

General Comments:

P5L25 The authors present their analysis averaged over a fixed latitude range (70-90N) rather than using a vortex following coordinate (e.g. by defining the edge of the vortex following Nash et al., 1996). Figure 12 in the manuscript shows the large zonal

variation in temperature and chemical fields, and highlights that the vortex is neither centred on the pole nor circular. I wonder what effect using a fairly large area average has on the results compared to averaging only within the vortex. While I do not feel it necessary to redo the analysis in any way, I would like to see a discussion on how using a fixed latitudinal average may affect the results of the paper compared to only considering airmasses within the vortex.

P3L22 While the authors have reference all the appropriate literature on the model configuration and description, and a detailed description of the EMAC model is not required, I would like to see further information on those parts of the model key to this paper. For example, section 2 should, in my mind, include a description of which PSC and aerosol types are included in the model, how sedimentation velocities are calculated, which heterogeneous reactions occur on aerosol surfaces, do uptake coefficients include temperature dependencies, etc. I feel this would significantly aid those not familiar with the EMAC CCM configuration.

Specific Comments:

P1L3 There is no need to capitalize polar stratospheric clouds here, and it should appear instead as it does in the Introduction (P2L10). However, in the Introduction it should read PCSs within the brackets.

P1L18 This is at odds with P7L32, where the authors state maximum ozone loss is 120 DU. While 2 ppmv is the maximum mixing ratio difference, 100 DU is more representative of the average loss over mid March, and does not represent the maximum column loss. This also applies to the conclusions (P11L19). I feel as well that it would be good to combine figures 7 and 8 so that total column differences appear below the deltaO3 plot in a single panel and the reader can compare the column loss with the altitudes at which this is occurring.

P3L4 I feel that having defined TNAT and PSC, these should be used consistently throughout the manuscript in place of NAT existence temperature and polar strato-

spheric clouds.

P7L12I feel deltaH2O should be defined in the text as deltaNOy and deltaO3 are. In fact, I feel each should be specifically defined in the text and figure captions (i.e. state deltaO3 = O3 – O3*).

P9L3 Is the Khosrawi et al. (2017) paper in prep, which it is in the reference list, or now published? If so this should be stated in the text. Further, if the paper is not yet available I do not feel that the reference should be included in this manuscript and reference to it removed (i.e. removed the sentence on P9L2-4. This also applies to the papers referenced on P10L27-28. Certainly they should say they are in prep if they are not yet published, and further if the findings of those studies are not key to this paper I do not feel they should be included.

P9L14 The simulations presented in the study are described as nudged in section 2. Therefore, surely any difference in temperature between the model and observations is a result of the nudged dataset and not the model. I feel saying 'temperatures as simulated with EMAC tend to be slightly warmer than measured outside the polar vortex' is misleading, as the temperature field is not being simulated freely. Presumably, in a free-running model the temperature biases would be significantly different.

P9L28 Without providing further information this a difficult conclusion to follow. Can the authors be sure that chlorine activation is not just too weak? The assertion in the manuscript reads as though the chlorine activation is correct, but petitioning between other active chlorine species is the cause of the low ClO values, indicating too high Cl, Cl2O2 etc. Can this be demonstrated by showing that ClONO2 and HCl are well simulated? Looking at these species should highlight the ability of the model to capture chlorine activation. Here also ClOx should be defined.

P11L4 The model simulations are nudged, and so is it still true that the EMAC model has weak downwards transport in this configuration? I would have thought that nudging the model ruled out dynamical factors as likely causes of any biases in chemical fields

when compared with observations.

P11L9 A further complication here is surely that if the fine-scale features are not present in the ECMWF dataset used for nudging then the model could never accurately capture these features. Perhaps a discussion on this and to what extent will this limit the ability of your future T255 model to reproduce this structure is warranted.

P12L1-3 This is true only for nudged configurations where the dynamics is accurately captured, and would not be true of free-running models. I feel this is an important point which should be made to caveat the conclusion.

Technical Corrections:

P11L29 ClOx should have a subscript x. Similarly subscripts should be used for NOy in Figure 4

Figure 1 I feel contours should be used consistently alongside the shading in the figures to aid with clarity, as is done in the top panel in Figure 1. This could be applied to all the pressure vs time plots.

Figure 13 It looks like there are zeros used for multiple contours in the top panels (ClO) in Figure 13, indicating the contour label does not have enough decimal places. This should be corrected. In a number of locations the grammar and sentence structure could be improved – I would encourage the authors to undertake another proof-read of the manuscript. The sentence on P9L30-32 should certainly be edited for clarity.

---

## Author Comment (AC1) · 14 Sep 2017

We thank reviewer 1 for the constructive, helpful criticism and the suggestion for revision. We followed the suggestions of reviewer 1 and revised the manuscript accordingly.

*General statement: This is a fine paper on an important topic that merits publication after revision. I do have several comments for the authors, delineated below. Important ones are marked with *.*

*\*1) The question of how denitrification and dehydration as such, versus a longer duration of cold temperatures into later parts of the spring season, have not been examined*

[Figure]

*quantitatively here. The authors should therefore avoid trying to make statements about how important denitrification and dehydration were (or would be) for the ozone loss. I suggest that the authors consider this point carefully in revision. I point out one place to make a change in text but I think there could well be others.*

We hope that with the changes we made in the frame of the revision all misleading sentences have been corrected.

*2) page 2, line 15. Please change 'ice' to 'water ice' here since some literature speaks of nitric acid ices. With this change, I don't think you need to say 'water ice' later in the text; doing it once is sufficient.*

We have changed "ice" to "water ice" as suggested.

*\*3) page 2, line 27. This statement makes a lot of assumptions that I don't think are merited. First, it ignores the literature on "denoxification", much of which suggests that denoxification later in the spring, when there is more sunlight, can be as important or more so in prolonging ozone loss provided temperatures are cold enough. Second (and related), I would argue that prolonging the ozone loss depends more on vortex stability and dynamics than it does on the degree of denitrification. Please add a discussion of these issues here, with appropriate references.*

The importance of denitrification for ozone loss was shown by e.g. Salawitch et al. 1993 and Rex et al. (1997). We added the missing references. For an additional discussion of denoxification and the importance of vortex stability we added the following paragraph in the introduction: *Another factor contributing to the severity of ozone destruction is the reduction of nitrogen ($NO_x = NO + NO_2$) via the conversion of $NO_x$ into $HNO_3$ on the surfaces of PSCs, the so-called denoxification. Denoxification becomes important if temperatures are continuously low during the course of the winter as is the case in the Antarctic (e. g. Waibel et al. 1999). It has been shown that polar vortex stability, chlorine activation and ozone loss tend to be greater with lower vortex temperatures (e. g. von Hobe et al., 2013). Therefore, it is not surprising that the most severe ozone loss ever observed in the*

[Figure]

*Arctic occurred in spring 2011, at the end of the most persistently cold Arctic winter in the stratosphere on record (Manney et al., 2011; Sinnhuber et al., 2011; Hommel et al., 2014)*

*4) page 4, line 28-29. I don't think these accuracy claims are true for MLS below 100 mb. Please check.*
We have checked this. In Livesey et al. (2017) useful range of Aura-MLS $O_3$ for scientific studies is given from 261-0.02 hPa. As stated in Livesey et al. (2017) there had been a high MLS v2.2 bias at 215 hPa observed in some comparisons versus certain ozonesonde and satellite datasets. These high biases, however, were reduced in versions v3.3x and v3.4x, with additional smaller reductions in the ozone values in v4.2x, the version that has been used in the present study. In addition, substantial oscillations that were present in the ozone profiles in previous versions have been ameliorated in v4.2x.

*5) page 5, line 2. Missing a word. Lowest retrieval level?*
Thanks for pointing this out. It indeed should read "lowest retrieval levels". This has been corrected.

*6) page 5, line 30. Reader needs a pointer ahead to indicate that you will define what you mean by 'unprecedented'. Add 'leading to unprecedented formation of ice PSCs (defined quantitatively below)'. . ..*
The sentence reads now: *Temperatures dropped during the first cold period (December to end of January) below the ice formation threshold temperatures (Manney et al., 2016) leading to unprecedented formation of ice PSCs as will be discussed in more detail below (see Fig. 2).*

*7) page 7, line 10. $2CH_4+H_2O$ isn't quite total hydrogen. I don't think it matters much for your purposes, but please have a look at LeTexier et al. (QJRMS, 1988) on this.*
This is correct, total hydrogen is properly defined as $2CH_4+H_2O+H_2$, but in the lower and middle stratosphere $H_2$ is constant and thus total hydrogen can in the lower/middle

stratosphere be derived from $2CH_4+H_2O$. We changed the text as follows to be more precise: *Dehydration from the EMAC simulation is derived by using total "stratospheric" hydrogen ($2CH_4+H_2O$) as substitute for a passive $H_2O$ tracer (e.g., Rinsland et al., 1996; Schiller et al., 1996). Molecular hydrogen ($H_2$) is nearly constant in the lower and middle stratosphere and can therefore be neglected in the calculation of total hydrogen. The quantity $2CH_4+H_20$ is generally constant in the stratosphere. However, slight deviations from this quasi-conserved quantity can be found at high latitudes during winter where transport of mesospheric air rich in molecular hydrogen and poor in water vapour and methane is brought into the upper stratosphere (e.g., Le Texier1988, Engel et al. 1996).*

*\*8) page 7, line 18, 19. Need to be more careful here. You could say something like 'The Arctic winter 2015/2016 had the greatest potential yet seen for record Arctic ozone loss if the vortex had remained stable (and temperatures had therefore remained cold) through late March'.*
We refer here to the results by Manney and Lawrence (2016) and changed the paragraph as follows to make this more clear: *The Arctic winter 2015/2016 appeared to have the greatest potential yet seen for record Arctic ozone loss (Manney and Lawrence, 2016). Temperatures in the Arctic lower stratosphere were at record lows from December 2015 to early February 2016 (Manney and Lawrence, 2016; Matthias et al., 2016). As was shown by Manney et al. (2016) ozone destruction began earlier and proceeded more rapidly than in 2010/2011, the winter that so far has been the one with the strongest observed ozone loss in the Arctic (Manney et al., 2011). That lower-stratospheric ozone loss did not reach the extent of that in spring 2011 was primarily due to a major final stratospheric warming in early March 2016 that led to a vortex split and a full breakdown of the vortex by early April (Manney et al., 2016)*

*\*9) page 9, line 30. Interesting – can you say something more about which other ClOx species are likely to be holding too much active chlorine? Cl2O2? ClONO2? Also, I don't think you can rule out that the activation is at the right time but just too weak? What is your*

*justification for ruling that out? Please clarify this here, as well as in other places where it is mentioned.*

It is correct that a possible explanation could also be that chlorine activation is just too weak. We unfortunately cannot rule out for sure what the cause of this discrepancy is. We know from other comparisons that there are also differences between the simulated and measured HCl and $ClONO_2$. Further, comparisons between different photolysis schemes performed by our colleagues at KIT (M. Sinnhuber and S. Versick) have revealed that the EMAC photolysis rates are too low at high solar zenith angles ($>90°$). The sentences have been changed as follows: *However, the enhancement of $ClO_x$ ($ClO_x=Cl+HOCl+2\cdot Cl_2+2\cdot Cl_2O_2$) in the EMAC simulation is found at the same time as in the Aura/MLS ClO observation, thus indicating that the later increase in ClO is not necessarily caused by the activation of chlorine being too late in the model simulation but could also be caused by the partitioning between the active chlorine species. In EMAC the photolysis rates are calculated with the submodel JVAL (Section 2.1). JVAL is part of the standard configuration of EMAC that was also used in the EMAC simulations contributing to the Chemistry Climate Initiative (CCMI, Jöckel et al., 2016) (note a similar configuration is used here apart from the resolution). An intercomparison of several photolysis scheme has shown that JVAL provides lower photolysis rates at very high solar zenith angles ($>90°$) for e.g. $Cl_2O_2$ than other schemes. Thus, the partitioning of chlorine containing species may be shifted for high solar zenith angles and thus could be the cause for the delay in the activation of ClO in the model simulation. However, to entirely rule out the cause for this difference further studies are necessary which however are beyond the scope of this study.* The sentence in the conclusion has been changed as follows: *Since the enhancement in modelled $ClO_x$ is found roughly at the same time as the increase in ClO observed by MLS, the disparity in the modelled and measured ClO may arise from chlorine activation being delayed in the model due to inaccuracies in the partitioning between chlorine species at high solar zenith angles.*
* * *
[Figure]

2017.

---

## Author Comment (AC2) · 14 Sep 2017

We thank reviewer 2 for the constructive, helpful criticism and the suggestion for revision. We followed the suggestions of reviewer 2 and revised the manuscript accordingly.

*Khosrawi et al. present a detailed analysis of polar processes occurring at high northern latitudes during the Arctic winter 2015/16. In particular, they compare simulations carried out with a nudged version of the EMAC CCM with a range of satellite and aircraft observations. The analysis presented in the paper is of high standard and explores an important and relevant topic within the scope of ACP and as such merits publication following revision. I have several comments the authors should address before publication:*

[Figure]

*General Comments:*

*P5L25 The authors present their analysis averaged over a fixed latitude range (70-90N) rather than using a vortex following coordinate (e.g. by defining the edge of the vortex following Nash et al., 1996). Figure 12 in the manuscript shows the large zonal variation in temperature and chemical fields, and highlights that the vortex is neither centred on the pole nor circular. I wonder what effect using a fairly large area average has on the results compared to averaging only within the vortex. While I do not feel it necessary to redo the analysis in any way, I would like to see a discussion on how using a fixed latitudinal average may affect the results of the paper compared to only considering airmasses within the vortex.*

In our analyses the usage of equivalent latitude is not mandatory since the separation between dynamics and chemistry is done by using the difference between the active (chemistry+dynamics) and the passive (dynamics only) tracer. However, in the frame of our analyses we have calculated ozone loss within an equivalent latitude band as well as within a geographic latitude band in order to quantify the differences in estimated ozone loss between the two approaches. Figure 1 and 2 in the supplement to this reply show ozone loss in mixing ratio and Dobson Units for both latitude and equivalent latitude. In terms of mixing ratios the result is almost the same (2.1 ppmv compared to 2.03 ppmv) while in Dobson Units the ozone loss on equivalent latitudes is approximately $10\%$ lower (117 DU compared to 103 DU). Figure 3 shows that there are slight differences between the $O_3$ column time series between latitude and equivalent latitude, but that our result remain the same, namely that in contrast to the other recent Arctic winters very low $O_3$ values are found in 2010/2011. We added the following text in section 3.4: *Note that, rather than employing a vortex following coordinate as e. g. equivalent latitudes, we have chosen to perform our analyses on a fixed geographic latitude band. Such an approach is justified here because the use of a passive tracer allows dynamical and chemical processes to be separated, thus faciliating the quantification of chemical ozone loss. On equivalent latitudes the same amount of ozone loss in terms of mixing ratio is derived while in terms of column loss ozone loss is $10\%$ less (103 DU).* In the

conclusion the following text has been added: *Note that we did not use equivalent latitudes here since separation between chemical and dynamical processes is achieved via the passive $O_3$ tracer. On equivalent latitudes the same amount of ozone loss in terms of mixing ratio is derived while in terms of column loss ozone loss is 10% less (103 DU)*

*P3L22 While the authors have reference all the appropriate literature on the model configuration and description, and a detailed description of the EMAC model is not required, I would like to see further information on those parts of the model key to this paper. For example, section 2 should, in my mind, include a description of which PSC and aerosol types are included in the model, how sedimentation velocities are calculated, which heterogeneous reactions occur on aerosol surfaces, do uptake coefficients include temperature dependencies, etc. I feel this would significantly aid those not familiar with the EMAC CCM configuration.*
We agree that it would be worthwile to provide more information on the parts of the model that are key to this paper. We added the following text briefly describing the PSC scheme and referring to Kirner et al. for more details: *The submodel MSBM simulates the number densities, mean radii and surface areas of sulphuric acid aerosols and liquid and solid polar stratospheric cloud particles. The formation of STS particles is calculated according to Carslaw et al. (1995) through the uptake of $HNO_3$ and $H_2O$ on the liquid binary sulphuric acid/water particles. Ice particles are assumed to form homogeneously at temperatures below $T_{ice}$. For the simulation of NAT particles the "kinetic growth NAT parameterisation" is used. The "kinetic" parameterisation is based on the growth and sedimentation algorithm given by Carslaw et al. (2002) and van den Broek et al. (2004). The vapour pressure over ice is calculated according to Marti and Mauersberger (1993) and the vapour pressure over NAT according to Hanson and Mauersberger (1988). NAT formation takes place as soon as a supercooling of 3 K below $T_{NAT}$ is reached. The sedimentation velocity of ice particles is calculated according to Waibel et al. (1997) and for NAT particles according to Carslaw et al. (2002). Eleven heterogeneous reactions that occur on the surfaces of liquid and solid PSC particles are considered. A comprehensive description of the submodel MSBM can be found in Kirner et al. (2011).*

*Specific Comments:*

*P1L3 There is no need to capitalize polar stratospheric clouds here, and it should appear instead as it does in the Introduction (P2L10). However, in the Introduction it should read PCSs within the brackets.*

This has been corrected.

*P1L18 This is at odds with P7L32, where the authors state maximum ozone loss is 120 DU. While 2 ppmv is the maximum mixing ratio difference, 100 DU is more representative of the average loss over mid March, and does not represent the maximum column loss. This also applies to the conclusions (P11L19).*

Thanks a lot for pointing this out. It should of course be the same amount of ozone loss in Dobson Units in all places of the paper. The exact amount is 117 DU. This has been corrected throughout the paper.

*I feel as well that it would be good to combine figures 7 and 8 so that total column differences appear below the $\Delta O_3$ plot in a single panel and the reader can compare the column loss with the altitudes at which this is occurring.*

We would prefer to not combine figures 7 and 8 since these figures show ozone loss in different units, namely DU and ppmv and combining these may be confusing for the reader. However, to make a comparison of these figures easier we adjusted the time axes of figure 8, so that both figures have the same time scale.

*P3L4 I feel that having defined $T_{\mathrm{NAT}}$ and PSC, these should be used consistently throughout the manuscript in place of NAT existence temperature and polar stratospheric clouds.*

We agree and now the abbreviations $T_{\mathrm{NAT}}$ and PSCs are used consistently throughout the manuscript.

*P7L12 I feel $\Delta H_2O$ should be defined in the text as $\Delta NO_y$ and $\Delta O_3$ are. In fact, I feel each should be specifically defined in the text and figure captions (i.e. state $\Delta O_3 = O_3 - O_3^*$).*

We followed the suggestion and each of the deltas are specifically defined in the text and figure captions.

*P9L3 Is the Khosrawi et al. (2017) paper in prep, which it is in the reference list, or now published? If so this should be stated in the text. Further, if the paper is not yet available I do not feel that the reference should be included in this manuscript and reference to it removed (i.e. removed the sentence on P9L2-4. This also applies to the papers referenced on P10L27-28. Certainly they should say they are in prep if they are not yet published, and further if the findings of those studies are not key to this paper I do not feel they should be included.*
We agree and removed the sentences referring to Sinnhuber et al. (2017), Braun et al. (2017) and Johansson et al. (2017) since these studies are not key to this paper and it is not yet clear when these papers will be submitted and published. We would like to keep the Khosrawi et al. (2017) reference since this paper is ready for submission, but kept on hold due to the new MIPAS PSC product which is not published yet. We anticipate to submit this paper in autum. Therefore, we changed the status in the reference list from "in preparation" to "to be submitted". Contrary to other journals as e.g. JGR, in the Copernicus journals the papers not published yet are listed with all other references in the reference list.

*P9L14 The simulations presented in the study are described as nudged in section 2. Therefore, surely any difference in temperature between the model and observations is a result of the nudged dataset and not the model. I feel saying 'temperatures as simulated with EMAC tend to be slightly warmer than measured outside the polar vortex' is misleading, as the temperature field is not being simulated freely. Presumably, in a free-running model the temperature biases would be significantly different.*
It is correct that the simulated temperatures in EMAC mainly reflect the temperature field of the meteorological analyses used for nudging the simulation. However, the EMAC temperatures and the temperatures from the ECMWF operational anlyses, used in our analyses for nudging, are not 100% identical although they are very similar.

The EMAC temperatures are not replaced by ECMWF operational temperatures, but the internally calculated EMAC temperatures are pushed toward the ECMWF operational analyses. Therefore, small differences between EMAC and ECMWF remain. We changed the sentence as follows: *Temperatures in EMAC (nudged towards ECMWF operational analyses) tend to be slightly warmer than measured outside the polar vortex.*

*P9L28 Without providing further information this a difficult conclusion to follow. Can the authors be sure that chlorine activation is not just too weak? The assertion in the manuscript reads as though the chlorine activation is correct, but petitioning between other active chlorine species is the cause of the low ClO values, indicating too high Cl, $Cl_2O_2$ etc. Can this be demonstrated by showing that $ClONO_2$ and HCl are well simulated? Looking at these species should highlight the ability of the model to capture chlorine activation. Here also $ClO_x$ should be defined.*

It is correct that a possible explanation could also be that chlorine activation is just too weak. We know from other comparisons that there are also differences between the simulated and measured HCl and $ClONO_2$. Further, comparisons between different photolysis schemes performed by our colleagues at KIT (M. Sinnhuber and S. Versick) have revealed that the EMAC photolysis rates are too low at high solar zenith angles (>90°). ClOx is now defined in the text and the discussion on the differences between EMAC and MLS in ClO has been changed as follows: *However, the enhancement of $ClO_x$ ($ClO_x$=Cl+HOCl+2·$Cl_2$+2·$Cl_2O_2$) in the EMAC simulation is found at the same time as in the Aura/MLS ClO observation, thus indicating that the later increase in ClO is not necessarily caused by the activation of chlorine being too late in the model simulation but could also be caused by the partitioning between the active chlorine species. In EMAC the photolysis rates are calculated with the submodel JVAL (Section 2.1). JVAL is part of the standard configuration of EMAC that was also used in the EMAC simulations contributing to the Chemistry-Climate Model Initiative (CCMI, Jöckel et al., 2016) (note a similar configuration is used here apart from the resolution). An intercomparison of several photolysis scheme has*

*shown that JVAL provides lower photolysis rates at very high solar zenith angles (>90°) for e.g. $Cl_2O_2$ than other schemes. Thus, the partitioning of chlorine containing species may be shifted for high solar zenith angles and thus could be the cause for the delay in the activation of ClO in the model simulation. However, to entirely rule out the cause for this difference further studies are necessary which however are beyond the scope of this study.*

*P11L4 The model simulations are nudged, and so is it still true that the EMAC model has weak downwards transport in this configuration? I would have thought that nudging the model ruled out dynamical factors as likely causes of any biases in chemical fields when compared with observations.*

Vertical winds are not nudged in EMAC, but divergence and vorticity are. In EMAC, the vertical wind is calculated with the help of these two parameters. Nevertheless, despite the nudging, the vertical transport is underestimated. The results are improved when a higher resolution is used, but the problem that the vertical transport is underestimated remains.

*P11L9 A further complication here is surely that if the fine-scale features are not present in the ECMWF dataset used for nudging then the model could never accurately capture these features. Perhaps a discussion on this and to what extent will this limit the ability of your future T255 model to reproduce this structure is warranted.*

The following text has been added to the last paragraph of section 4.2 to discuss this:
*However, it should be kept in mind that a good agreement between model simulations and observations can only be obtained if the model simulations are nudged towards meteorological analyses. It can be expected that comparison with free running model simulations would show larger differences. Further, the results are also limited by the accuracy of the meteorological analyses, e.g. resolving small-scale temperature fluctuations and mountain waves will still be problematic even when a T255 resolution is used.*

*P12L1-3 This is true only for nudged configurations where the dynamics is accurately*

*captured, and would not be true of free-running models. I feel this is an important point which should be made to caveat the conclusion.*

To be more clear on this point we mention now at several places in the conclusions that a nudged EMAC simulation was used.

*Technical Corrections:*

*P11L29 $ClO_x$ should have a subscript x. Similarly subscripts should be used for $NO_y$ in Figure 4.*

Thanks for pointing this out. This has been corrected.

*Figure 1 I feel contours should be used consistently alongside the shading in the figures to aid with clarity, as is done in the top panel in Figure 1. This could be applied to all the pressure vs time plots.*

We have tried this, but found that the addition of extra contours make the Delta and PSC plots too cluttered and thus harder to interpret.

*Figure 13 It looks like there are zeros used for multiple contours in the top panels (ClO) in Figure 13, indicating the contour label does not have enough decimal places. This should be corrected.*

Thanks for pointing this out. The figure has been corrected.

*In a number of locations the grammar and sentence structure could be improved - I would encourage the authors to undertake another proof-read of the manuscript. The sentence on P9L30-32 should certainly be edited for clarity.*

We have performed another proof-read of the manuscript and hope that everything is correct now.

Figures are provided as supplement to this reply.

Please also note the supplement to this comment:
https://www.atmos-chem-phys-discuss.net/acp-2017-503/acp-2017-503-AC2-supplement.pdf

**Supplement:**

**Reply to Referee 2 Comments**

Manuscript-No: acp-2017-503

**Denitrification, dehydration and ozone loss during the Arctic winter 2015/2016**

[Figure]

Figure 1: Ozone loss from EMAC T106L90 simulation at 34 hPa for the Arctic winter 2015/2016. Ozone loss has been derived from the difference between the active tracer $O_3$ and the passive tracer $O_3^*$ ($\Delta O_3 = O_3 - O_3^*$). Top: average over 70-90°N latitude, bottom: average over 70-90°N equivalent latitude.

[Figure]

Figure 2: Total column ozone loss derived from the EMAC T106L90 simulation. Ozone loss has been derived from the difference between the active tracer $O_3$ at the passive tracer $O_3^*$ ($\Delta O_3 = O_3 - O_3^*$). Top: average over 70-90°N latitude, bottom: average over 70-90°N equivalent latitude.

[Figure]

Figure 3: Ozone column time series for the Arctic winters 2009/2010 (blue), 2010/2011 (green), 2013/2014 (red) and 2015/2016 (magenta) averaged over 60-90°N latitude (top) and 60-90°N equivalent latitude (bottom). Results from the EMAC T42L90 simulation are shown.

---

## Author Response (ED1)

**Reply to Referee 1 Comments**

**Manuscript-No: acp-2017-503**

**Denitrification, dehydration and ozone loss during the Arctic winter 2015/2016**

We thank reviewer 1 for the constructive, helpful criticism and the suggestion for revision. We followed the suggestions of reviewer 1 and revised the manuscript accordingly.

*General statement: This is a fine paper on an important topic that merits publication after revision. I do have several comments for the authors, delineated below. Important ones are marked with \*.*

*\*1) The question of how denitrification and dehydration as such, versus a longer duration of cold temperatures into later parts of the spring season, have not been examined quantitatively here. The authors should therefore avoid trying to make statements about how important denitrification and dehydration were (or would be) for the ozone loss. I suggest that the authors consider this point carefully in revision. I point out one place to make a change in text but I think there could well be others.*
We hope that with the changes we made in the frame of the revision all misleading sentences have been corrected.

*2) page 2, line 15. Please change ice to water ice here since some literature speaks of nitric acid ices. With this change, I dont think you need to say water ice later in the text; doing it once is sufficient.*
We have changed "ice" to "water ice" as suggested.

*\*3) page 2, line 27. This statement makes a lot of assumptions that I don't think are merited. First, it ignores the literature on "denoxification", much of which suggests that denoxification later in the spring, when there is more sunlight, can be as important or more so in prolonging ozone loss provided temperatures are cold enough. Second (and related), I would argue that prolonging the ozone loss depends more on vortex stability and dynamics than it does on the degree of denitrification. Please add a discussion of these issues here, with appropriate references.*
The importance of denitrification for ozone loss was shown by e.g. Salawitch et al. 1993 and Rex et al. (1997). We added the missing references. For an additional discussion of denoxification and the importance of vortex stability we added the following paragraph in the introduction: *Another factor contributing to the severity of ozone destruction is the reduction of nitrogen ($NO_x=NO+NO_2$) via the conversion of $NO_x$ into $HNO_3$ on the surfaces of PSCs, the so-called denoxification. Denoxification becomes important if*

*temperatures are continuously low during the course of the winter as is the
case in the Antarctic (e.g. Waibel et al. 1999). It has been shown that
polar vortex stability, chlorine activation and ozone loss tend to be greater
with lower vortex temperatures (e.g. von Hobe et al., 2013). Therefore, it
is not surprising that the most severe ozone loss ever observed in the Arc-
tic occurred in spring 2011, at the end of the most persistently cold Arctic
winter in the stratosphere on record (Manney et al., 2011; Sinnhuber et al.,
2011; Hommel et al., 2014)*

*4) page 4, line 28-29. I don't think these accuracy claims are true for MLS
below 100 mb. Please check.*
We have checked this. In Livesey et al. (2017) useful range of Aura-MLS $O_3$
for scientific studies is given from 261-0.02 hPa. As stated in Livesey et al.
(2017) there had been a high MLS v2.2 bias at 215 hPa observed in some
comparisons versus certain ozonesonde and satellite datasets. These high
biases, however, were reduced in versions v3.3x and v3.4x, with additional
smaller reductions in the ozone values in v4.2x, the version that has been
used in the present study. In addition, substantial oscillations that were
present in the ozone profiles in previous versions have been ameliorated in
v4.2x.

*5) page 5, line 2. Missing a word. Lowest retrieval level?*
Thanks for pointing this out. It indeed should read "lowest retrieval levels".
This has been corrected.

*6) page 5, line 30. Reader needs a pointer ahead to indicate that you will
define what you mean by unprecedented. Add leading to unprecedented for-
mation of ice PSCs (defined quantitatively below). . ..*
The sentence reads now: *Temperatures dropped during the first cold period
(December to end of January) below the ice formation threshold tempera-
tures (Manney et al., 2016) leading to unprecedented formation of ice PSCs
as will be discussed in more detail below (see Fig. 2).*

*7) page 7, line 10. $2CH_4+H_2O$ isn't quite total hydrogen. I don't think
it matters much for your purposes, but please have a look at LeTexier et al.
(QJRMS, 1988) on this.*
This is correct, total hydrogen is properly defined as $2CH_4+H_2O+H_2$, but in
the lower and middle stratosphere $H_2$ is constant and thus total hydrogen can
in the lower/middle stratosphere be derived from $2CH_4+H_2O$. We changed
the text as follows to be more precise: *Dehydration from the EMAC sim-
ulation is derived by using total "stratospheric" hydrogen ($2CH_4+H_2O$) as
substitute for a passive $H_2O$ tracer (e.g., Rinsland et al., 1996; Schiller et al.,
1996). Molecular hydrogen ($H_2$) is nearly constant in the lower and middle
stratosphere and can therefore be neglected in the calculation of total hy-*

drogen. The quantity $2CH_4+H_20$ is generally constant in the stratosphere. However, slight deviations from this quasi-conserved quantity can be found at high latitudes during winter where transport of mesospheric air rich in molecular hydrogen and poor in water vapour and methane is brought into the upper stratosphere (e.g., Le Texier1988, Engel et al. 1996).

*8) page 7, line 18, 19. Need to be more careful here. You could say something like The Arctic winter 2015/2016 had the greatest potential yet seen for record Arctic ozone loss if the vortex had remained stable (and temperatures had therefore remained cold) through late March.
We refer here to the results by Manney and Lawrence (2016) and changed the paragraph as follows to make this more clear: *The Arctic winter 2015/2016 appeared to have the greatest potential yet seen for record Arctic ozone loss (Manney and Lawrence, 2016). Temperatures in the Arctic lower stratosphere were at record lows from December 2015 to early February 2016 (Manney and Lawrence, 2016; Matthias et al., 2016). As was shown by Manney et al. (2016) ozone destruction began earlier and proceeded more rapidly than in 2010/2011, the winter that so far has been the one with the strongest observed ozone loss in the Arctic (Manney et al., 2011). That lower-stratospheric ozone loss did not reach the extent of that in spring 2011 was primarily due to a major final stratospheric warming in early March 2016 that led to a vortex split and a full breakdown of the vortex by early April (Manney et al., 2016)*

*9) page 9, line 30. Interesting can you say something more about which other ClOx species are likely to be holding too much active chlorine? Cl2O2? ClONO2? Also, I dont think you can rule out that the activation is at the right time but just too weak? What is your justification for ruling that out? Please clarify this here, as well as in other places where it is mentioned.
It is correct that a possible explanation could also be that chlorine activation is just too weak. We unfortunately cannot rule out for sure what the cause of this discrepancy is. We know from other comparisons that there are also differences between the simulated and measured HCl and $ClONO_2$. Further, comparisons between different photolysis schemes performed by our colleagues at KIT (M. Sinnhuber and S. Versick) have revealed that the EMAC photolysis rates are too low at high solar zenith angles (¿90°). The sentences have been changed as follows: *However, the enhancement of $ClO_x$ ($ClO_x$=$Cl$+$HOCl$+$2\cdot Cl_2$+$2\cdot Cl_2O_2$) in the EMAC simulation is found at the same time as in the Aura/MLS ClO observation, thus indicating that the later increase in ClO is not necessarily caused by the activation of chlorine being too late in the model simulation but could also be caused by the partitioning between the active chlorine species. In EMAC the photolysis rates are calculated with the submodel JVAL (Section 2.1). JVAL is part of the standard configuration of EMAC that was also used in the EMAC*

*simulations contributing to the Chemistry Climate Initiative (CCMI, Jöckel et al., 2016) (note a similar configuration is used here apart from the resolution). An intercomparison of several photolysis scheme has shown that JVAL provides lower photolysis rates at very high solar zenith angles (¿90°) for e.g. $Cl_2O_2$ than other schemes. Thus, the partitioning of chlorine containing species may be shifted for high solar zenith angles and thus could be the cause for the delay in the activation of ClO in the model simulation. However, to entirely rule out the cause for this difference further studies are necessary which however are beyond the scope of this study. The sentence in the conclusion has been changed as follows: Since the enhancement in modelled $ClO_x$ is found roughly at the same time as the increase in ClO observed by MLS, the disparity in the modelled and measured ClO may arise from chlorine activation being delayed in the model due to inaccuracies in the partitioning between chlorine species at high solar zenith angles.*

**Reply to Referee 2 Comments**

**Manuscript-No: acp-2017-503**

**Denitrification, dehydration and ozone loss during the Arctic winter 2015/2016**

We thank reviewer 2 for the constructive, helpful criticism and the suggestion for revision. We followed the suggestions of reviewer 2 and revised the manuscript accordingly.

*Khosrawi et al. present a detailed analysis of polar processes occurring at high northern latitudes during the Arctic winter 2015/16. In particular, they compare simulations carried out with a nudged version of the EMAC CCM with a range of satellite and aircraft observations. The analysis presented in the paper is of high standard and explores an important and relevant topic within the scope of ACP and as such merits publication following revision. I have several comments the authors should address before publication:*

*General Comments:*
*P5L25 The authors present their analysis averaged over a fixed latitude range (70-90N) rather than using a vortex following coordinate (e.g. by defining the edge of the vortex following Nash et al., 1996). Figure 12 in the manuscript shows the large zonal variation in temperature and chemical fields, and highlights that the vortex is neither centred on the pole nor circular. I wonder what effect using a fairly large area average has on the results compared to averaging only within the vortex. While I do not feel it necessary to redo the analysis in any way, I would like to see a discussion on how using a fixed latitudinal average may affect the results of the paper compared to only considering airmasses within the vortex.*

In our analyses the usage of equivalent latitude is not mandatory since the separation between dynamics and chemistry is done by using the difference between the active (chemistry+dynamics) and the passive (dynamics only) tracer. However, in the frame of our analyses we have calculated ozone loss within an equivalent latitude band as well as within a geographic latitude band in order to quantify the differences in estimated ozone loss between the two approaches. Figure 1 and 2 in this reply show ozone loss in mixing ratio and Dobson Units for both latitude and equivalent latitude. In terms of mixing ratios the result is almost the same (2.1 ppmv compared to 2.03 ppmv) while in Dobson Units the ozone loss on equivalent latitudes is approximately 10% lower (117 DU compared to 103 DU). Figure 3 shows that there are slight differences between the $O_3$ column time series between latitude and equivalent latitude, but that our result remain the same, namely that in contrast to the other recent Arctic winters very low $O_3$ values are found in 2010/2011. We added the following text in section 3.4: *Note that,*

*rather than employing a vortex following coordinate as e. g. equivalent latitudes, we have chosen to perform our analyses on a fixed geographic latitude band. Such an approach is justified here because the use of a passive tracer allows dynamical and chemical processes to be separated, thus faciliating the quantification of chemical ozone loss. On equivalent latitudes the same amount of ozone loss in terms of mixing ratio is derived while in terms of column loss ozone loss is 10 % less (103 DU).* In the conclusion the following text has been added: *Note that we did not use equivalent latitudes here since separation between chemical and dynamical processes is achieved via the passive $O_3$ tracer. On equivalent latitudes the same amount of ozone loss in terms of mixing ratio is derived while in terms of column loss ozone loss is 10 % less (103 DU)*

*P3L22 While the authors have reference all the appropriate literature on the model configuration and description, and a detailed description of the EMAC model is not required, I would like to see further information on those parts of the model key to this paper. For example, section 2 should, in my mind, include a description of which PSC and aerosol types are included in the model, how sedimentation velocities are calculated, which heterogeneous reactions occur on aerosol surfaces, do uptake coefficients include temperature dependencies, etc. I feel this would significantly aid those not familiar with the EMAC CCM configuration.*

We agree that it would be worthwile to provide more information on the parts of the model that are key to this paper. We added the following text briefly describing the PSC scheme and referring to Kirner et al. for more details: *The submodel MSBM simulates the number densities, mean radii and surface areas of sulphuric acid aerosols and liquid and solid polar stratospheric cloud particles. The formation of STS particles is calculated according to Carslaw et al. (1995) through the uptake of $HNO_3$ and $H_2O$ on the liquid binary sulphuric acid/water particles. Ice particles are assumed to form homogeneously at temperatures below $T_{ice}$. For the simulation of NAT particles the "kinetic growth NAT parameterisation" is used. The "kinetic" parameterisation is based on the growth and sedimentation algorithm given by Carslaw et al. (2002) and van den Broek et al. (2004). The vapour pressure over ice is calculated according to Marti and Mauersberger (1993) and the vapour pressure over NAT according to Hanson and Mauersberger (1988). NAT formation takes place as soon as a supercooling of 3 K below $T_{NAT}$ is reached. The sedimentation velocity of ice particles is calculated according to Waibel et al. (1997) and for NAT particles according to Carslaw et al. (2002). Eleven heterogeneous reactions that occur on the surfaces of liquid and solid PSC particles are considered. A comprehensive description of the submodel MSBM can be found in Kirner et al. (2011).*

*Specific Comments:*

*P1L3 There is no need to capitalize polar stratospheric clouds here, and it should appear instead as it does in the Introduction (P2L10). However, in the Introduction it should read PCSs within the brackets.*
This has been corrected.

*P1L18 This is at odds with P7L32, where the authors state maximum ozone loss is 120 DU. While 2 ppmv is the maximum mixing ratio difference, 100 DU is more representative of the average loss over mid March, and does not represent the maximum column loss. This also applies to the conclusions (P11L19).*
Thanks a lot for pointing this out. It should of course be the same amount of ozone loss in Dobson Units in all places of the paper. The exact amount is 117 DU. This has been corrected throughout the paper.

*I feel as well that it would be good to combine figures 7 and 8 so that total column differences appear below the $\Delta O_3$ plot in a single panel and the reader can compare the column loss with the altitudes at which this is occurring.*
We would prefer to not combine figures 7 and 8 since these figures show ozone loss in different units, namely DU and ppmv and combining these may be confusing for the reader. However, to make a comparison of these figures easier we adjusted the time axes of figure 8, so that both figures have the same time scale.

*P3L4 I feel that having defined $T_{\mathrm{NAT}}$ and PSC, these should be used consistently throughout the manuscript in place of NAT existence temperature and polar stratospheric clouds.*
We agree and now the abbreviations $T_{\mathrm{NAT}}$ and PSCs are used consistently throughout the manuscript.

*P7L12 I feel $\Delta H_2O$ should be defined in the text as $\Delta NO_y$ and $\Delta O_3$ are. In fact, I feel each should be specifically defined in the text and figure captions (i.e. state $\Delta O_3 = O_3 - O_3^*$).*
We followed the suggestion and each of the deltas are specifically defined in the text and figure captions.

*P9L3 Is the Khosrawi et al. (2017) paper in prep, which it is in the reference list, or now published? If so this should be stated in the text. Further, if the paper is not yet available I do not feel that the reference should be included in this manuscript and reference to it removed (i.e. removed the sentence on P9L2-4. This also applies to the papers referenced on P10L27-28. Certainly they should say they are in prep if they are not yet published, and further if the findings of those studies are not key to this paper I do not feel they should be included.*

We agree and removed the sentences referring to Sinnhuber et al. (2017), Braun et al. (2017) and Johansson et al. (2017) since these studies are not key to this paper and it is not yet clear when these papers will be submitted and published. We would like to keep the Khosrawi et al. (2017) reference since this paper is ready for submission, but kept on hold due to the new MIPAS PSC product which is not published yet. We anticipate to submit this paper in autum. Therefore, we changed the status in the reference list from "in preparation" to "to be submitted". Contrary to other journals as e.g. JGR, in the Copernicus journals the papers not published yet are listed with all other references in the reference list.

*P9L14 The simulations presented in the study are described as nudged in section 2. Therefore, surely any difference in temperature between the model and observations is a result of the nudged dataset and not the model. I feel saying temperatures as simulated with EMAC tend to be slightly warmer than measured outside the polar vortex is misleading, as the temperature field is not being simulated freely. Presumably, in a free-running model the temperature biases would be significantly different.*
It is correct that the simulated temperatures in EMAC mainly reflect the temperature field of the meteorological analyses used for nudging the simulation. However, the EMAC temperatures and the temperatures from the ECMWF operational anlyses, used in our analyses for nudging, are not 100% identical although they are very similar. The EMAC temperatures are not replaced by ECMWF operational temperatures, but the internally calculated EMAC temperatures are pushed toward the ECMWF operational analyses. Therefore, small differences between EMAC and ECMWF remain. We changed the sentence as follows: *Temperatures in EMAC (nudged towards ECMWF operational analyses) tend to be slightly warmer than measured outside the polar vortex.*

*P9L28 Without providing further information this a difficult conclusion to follow. Can the authors be sure that chlorine activation is not just too weak? The assertion in the manuscript reads as though the chlorine activation is correct, but petitioning between other active chlorine species is the cause of the low ClO values, indicating too high Cl, $Cl_2O_2$ etc. Can this be demonstrated by showing that $ClONO_2$ and HCl are well simulated? Looking at these species should highlight the ability of the model to capture chlorine activation. Here also $ClO_x$ should be defined.*
It is correct that a possible explanation could also be that chlorine activation is just too weak. We know from other comparisons that there are also differences between the simulated and measured HCl and $ClONO_2$. Further, comparisons between different photolysis schemes performed by our colleagues at KIT (M. Sinnhuber and S. Versick) have revealed that the EMAC photolysis rates are too low at high solar zenith angles (¿90°). ClOx is now

defined in the text and the discussion on the differences between EMAC and MLS in ClO has been changed as follows: *However, the enhancement of $ClO_x$ ($ClO_x$=Cl+HOCl+2·$Cl_2$+2·$Cl_2O_2$) in the EMAC simulation is found at the same time as in the Aura/MLS ClO observation, thus indicating that the later increase in ClO is not necessarily caused by the activation of chlorine being too late in the model simulation but could also be caused by the partitioning between the active chlorine species. In EMAC the photolysis rates are calculated with the submodel JVAL (Section 2.1). JVAL is part of the standard configuration of EMAC that was also used in the EMAC simulations contributing to the Chemistry-Climate Model Initiative (CCMI, Jöckel et al., 2016) (note a similar configuration is used here apart from the resolution). An intercomparison of several photolysis scheme has shown that JVAL provides lower photolysis rates at very high solar zenith angles (¿90°) for e.g. $Cl_2O_2$ than other schemes. Thus, the partitioning of chlorine containing species may be shifted for high solar zenith angles and thus could be the cause for the delay in the activation of ClO in the model simulation. However, to entirely rule out the cause for this difference further studies are necessary which however are beyond the scope of this study.*

*P11L4 The model simulations are nudged, and so is it still true that the EMAC model has weak downwards transport in this configuration? I would have thought that nudging the model ruled out dynamical factors as likely causes of any biases in chemical fields when compared with observations.*
Vertical winds are not nudged in EMAC, but divergence and vorticity are. In EMAC, the vertical wind is calculated with the help of these two parameters. Nevertheless, despite the nudging, the vertical transport is underestimated. The results are improved when a higher resolution is used, but the problem that the vertical transport is underestimated remains.

*P11L9 A further complication here is surely that if the fine-scale features are not present in the ECMWF dataset used for nudging then the model could never accurately capture these features. Perhaps a discussion on this and to what extent will this limit the ability of your future T255 model to reproduce this structure is warranted.*
The following text has been added to the last paragraph of section 4.2 to discuss this: *However, it should be kept in mind that a good agreement between model simulations and observations can only be obtained if the model simulations are nudged towards meteorological analyses. It can be expected that comparison with free running model simulations would show larger differences. Further, the results are also limited by the accuracy of the meteorological analyses, e.g. resolving small-scale temperature fluctuations and mountain waves will still be problematic even when a T255 resolution is used.*

*P12L1-3 This is true only for nudged configurations where the dynamics is accurately captured, and would not be true of free-running models. I feel this is an important point which should be made to caveat the conclusion.*
To be more clear on this point we mention now at several places in the conclusions that a nudged EMAC simulation was used.

*Technical Corrections:*
*P11L29 $ClO_x$ should have a subscript x. Similarly subscripts should be used for $NO_y$ in Figure 4.*
Thanks for pointing this out. This has been corrected.

*Figure 1 I feel contours should be used consistently alongside the shading in the figures to aid with clarity, as is done in the top panel in Figure 1. This could be applied to all the pressure vs time plots.*
We have tried this, but found that the addition of extra contours make the Delta and PSC plots too cluttered and thus harder to interpret.

*Figure 13 It looks like there are zeros used for multiple contours in the top panels (ClO) in Figure 13, indicating the contour label does not have enough decimal places. This should be corrected.*
Thanks for pointing this out. The figure has been corrected.

*In a number of locations the grammar and sentence structure could be improved - I would encourage the authors to undertake another proof-read of the manuscript. The sentence on P9L30-32 should certainly be edited for clarity.*
We have performed another proof-read of the manuscript and hope that everything is correct now.

[Figure]

Figure 1: Ozone loss from EMAC T106L90 simulation at 34 hPa for the Arctic winter 2015/2016. Ozone loss has been derived from the difference between the active tracer $O_3$ and the passive tracer $O_3^*$ ($\Delta O_3 = O_3 - O_3^*$). Top: average over 70-90°N latitude, bottom: average over 70-90°N equivalent latitude.

[Figure]

Figure 2: Total column ozone loss derived from the EMAC T106L90 simulation. Ozone 
[revised manuscript text omitted]
_{\text{liq}}$ (µm$^2$/cm$^3$), $A_{\text{NAT}}$ ($10^{-3}$ µm$^2$/cm$^3$) and $A_{\text{ice}}$ ($10^{-1}$ µm$^2$/cm$^3$).

[Figure]

**Figure 3.** Distribution of $HNO_3$ as simulated with EMAC T106L90 at 52 hPa on certain dates between 24 December 2015 and 12 February 2016.

[Figure]

**Figure 4.** Redistribution of  NO$_y$ (ΔNO$_y$) simulated with EMAC T106L90 (difference of  NO$_y$ and the passive tracer NO$_y^*$ (ΔNO$_y$=NO$_y$-NO$_y^*$), averaged over 70-90°N).

**EMAC T106L90 H$_2$O at 52 hPa**

[Figure]

**Figure 5.** Distribution of H$_2$O as simulated with EMAC T106L90 at 52hPa on certain dates between 24 December 2015 and 12 February 2016.

[Figure]

**Figure 6.** Redistribution of $H_2O$ ($\Delta H_2O$) as simulated with EMAC T106L90 at northern high latitudes (70-90°N) as function of pressure during the Arctic winter 2015/2016 (difference of total hydrogen (2$CH_4$+$H_2O$ )at time $t$ and total hydrogen at time $t_0$=1 December ($\Delta H_2O$=(2$CH_4$+$H_2O$)($t$)-(2$CH_4$+$H_2O$)($t_0$)).

[Figure]

**Figure 7.** Ozone loss $(\Delta O_3)$ as simulated with EMAC T106L90 (difference of $O_3$ and the passive tracer $O_3^*$ $(\Delta O_3 = O_3 - O_3^*)$, averaged over 70-90°N) as function of time and pressure for the Arctic winter 2015/2016.

[Figure]

**Figure 8.** Total column ozone loss ($\Delta O_3$) from EMAC T106L90 (70-90°N) for the Arctic winter 2015/2016. Total column loss has been derived from the difference between the active tracer $O_3$ and the passive tracer $O_3^*$ ($\Delta O_3 = O_3 - O_3^*$).

[Figure]

**Figure 9.** Total ozone column (March monthly mean) from EMAC for the years 2010-2016 (Results from the EMAC T42L90 Simulation are shown here).

[Figure]

**Figure 10.** Ozone ($O_3$) column time series for the Arctic winters 2009/2010 (blue), 2010/2011 (green), 2013/2014 (red) and 2015/2016 (magenta) averaged over 60-90°N (Results from the EMAC T42L90 Simulation are shown here).

[Figure]

**Figure 11.** Tracer time series of $HNO_3$ and $H_2O$ for the Arctic winters 2009/2010 (blue), 2010/2011 (green), 2013/2014 (red) and 2015/2016 (magenta) at 48 hPa averaged over 70-90°N (Result from the T42L90 Simulation are shown here).

[Figure]

**Figure 12.** Temperature, HNO$_3$, O$_3$ distribution measured by Aura/MLS (left) and simulated by EMAC T106L90 (right) at ∼50 hPa on 15 January 2016.

[Figure]

**Figure 13.** Temporal evolution of daily mean ClO, HNO$_3$ and O$_3$ at northern high latitudes (averaged over 70-90°N) as function of pressure as observed by Aura/MLS (left) and simulated by EMAC T106L90 (right) for the Arctic winter 2015/2016 (EMAC SORBIT output used).

[Figure]

[Figure]

**Figure 14.** Time series of $HNO_3$ and $O_3$ from Aura/MLS measurements (grey) and from the EMAC T42L90 (blue), EMAC T106L90 (green) at ∼50 hPa averaged over 70-90°N (EMAC SORBIT output used).

[Figure]

**Figure 15.** GLORIA HNO$_3$ and O$_3$ observations during flight 21 on 18 March 2016 (left) and EMAC T106L90 output along the flight track (right).

---

## Author Response (AR2)

**Reply to Co-Editor Comments**

**Manuscript-No: acpd-2017-503**

**Denitrification, dehydration and ozone loss during the Arctic winter 2015/2016**

We thank the co-editor for carefully reading the manuscript and providing technical corrections before the paper is published in ACP.

[revised manuscript text omitted]